# GhostKnockoff inference empowers identification of putative causal variants in genome-wide association studies

Zihuai He [1,2] ✉, Linxi Liu[3], Michael E. Belloy[1], Yann Le Guen [1,4], Aaron Sossin[5], Xiaoxia Liu[1], Xinran Qi[1], Shiyang Ma [6], Prashnna K. Gyawali[1], Tony Wyss-Coray [1], Hua Tang [7], Chiara Sabatti[5], Emmanuel Candès[8,9], Michael D. Greicius[1] & Iuliana Ionita-Laza[6]

Recent advances in genome sequencing and imputation technologies provide an exciting opportunity to comprehensively study the contribution of genetic variants to complex phenotypes. However, our ability to translate genetic discoveries into mechanistic insights remains limited at this point. In this paper, we propose an efficient knockoff-based method, GhostKnockoff, for genome-wide association studies (GWAS) that leads to improved power and ability to prioritize putative causal variants relative to conventional GWAS approaches. The method requires only Z-scores from conventional GWAS and hence can be easily applied to enhance existing and future studies. The method can also be applied to meta-analysis of multiple GWAS allowing for arbitrary sample overlap. We demonstrate its performance using empirical simulations and two applications: (1) a meta-analysis for Alzheimer's disease comprising nine overlapping large-scale GWAS, whole-exome and whole-genome sequencing studies and (2) analysis of 1403 binary phenotypes from the UK Biobank data in 408,961 samples of European ancestry. Our results demonstrate that GhostKnockoff can identify putatively functional variants with weaker statistical effects that are missed by conventional association tests.

Recent advances in genome sequencing technologies and improvement in genotype imputation accuracy enable large-scale genetic studies with hundreds of thousands of samples and tens of millions of variants. The ultimate goal of such studies is to provide a credible set of putative causal variants that could lead to novel targets for the development of genomic-driven medicine. However, our ability to identify causal genetic variants and to translate genetic discoveries into mechanistic insights and drug targets remains limited at this point[1]. Conventional genome-wide association studies (GWAS) are based on marginal association models that regress an outcome of interest on a single genetic variant at a time, using Bonferroni correction for the number of independent tests to control the family-wise error rate (FWER). Although this approach based on marginal test statistics has successfully discovered many disease-associated variants, its statistical power can be suboptimal, and it often identifies proxy variants that are only correlated with the true causal variants[2].

[1]Department of Neurology and Neurological Sciences, Stanford University, Stanford, CA 94305, USA. [2]Quantitative Sciences Unit, Department of Medicine, Stanford University, Stanford, CA 94305, USA. [3]Department of Statistics, University of Pittsburgh, Pittsburgh, PA 15260, USA. [4]Institut du Cerveau - Paris Brain Institute - ICM, Paris 75013, France. [5]Department of Biomedical Data Science, Stanford University, Stanford, CA 94305, USA. [6]Department of Biostatistics, Columbia University, New York, NY 10032, USA. [7]Department of Genetics, Stanford University, Stanford, CA 94305, USA. [8]Department of Statistics, Stanford University, Stanford, CA 94305, USA. [9]Department of Mathematics, Stanford University, Stanford, CA 94305, USA. ✉e-mail: zihuai@stanford.edu

Multiple lines of genetic research suggest that small effect risk loci that currently lie below the genome-wide significance threshold even in large GWAS can be informative to understand complex phenotypes. First, the widely used polygenic models for complex phenotypes are based on the idea that heritability can be explained by a large number of loci, each with small or infinitesimal effects[3,4]. Second, because of the polygenic nature of complex traits, inclusion of loci of small effects improves the predictive power of polygenic risk scores (PRS) for many traits[5]. Third, although the effect sizes of such loci are small in populations, they can still provide important biological insights. In particular, their effect on molecular phenotypes can be large and they can lead to effective drug targets (e.g. statins)[6]. However, small effect loci are difficult to distinguish from noisy loci, especially with suboptimal marginal association tests commonly used in GWAS.

Knockoff inference is a recently proposed statistical framework for variable selection in high-dimensional settings[7,8]. Unlike marginal association testing in GWAS, the knockoff-based inference performs genome-wide conditional tests that account for linkage disequilibrium (LD) thereby reducing false positive findings due to LD confounding. It provides rigorous control of the false discovery rate (FDR), i.e. the expected proportion of discoveries which correspond to truly null hypotheses. The idea of the knockoff-based inference is to generate synthetic, noisy copies (knockoffs) of the original genetic variants that resemble the true variants in terms of preserving correlations but are conditionally independent of the disease phenotype given the original variants. The knockoffs serve as negative controls for the conditional tests to select significant genetic risk loci and to attenuate the confounding effect of LD. Unlike the conventional Benjamini-Hochberg procedure that does not account for LD, the knockoff framework appropriately accounts for arbitrary correlations among the conditional tests while guaranteeing control of the FDR[9].

Several knockoff-based methods have already been proposed for genetic studies including Candès et al.[7], Sesia et al.[10], Sesia et al.[11], He et al.[12] and Sesia et al.[13]. These showed that controlling FDR can be more powerful to identify causal variants with weaker effect sizes relative to conventional GWAS, under the assumption of a polygenic model. In particular, they demonstrated that the variants identified by the knockoff inference are more likely to be the causal ones. Despite these appealing features, individual level data needed for the knockoff generation are often not available in large meta-analyses GWAS; instead, summary statistics that do not contain individual identifiable information are usually available. Furthermore, the high computational and memory cost needed to generate individual data knockoffs represents a major bottleneck to achieve its full potential. Finally, unlike for the traditional GWAS, there is currently a lack of standardized, efficient pipelines to facilitate the application of knockoff-based inference to genetic studies.

In this paper, we propose a novel method, *GhostKnockoff*, to allow efficient knockoff-based inference using freely available GWAS summary statistics for enhanced locus discovery and genome-wide prioritization of causal variants. Methodologically, we show that for the conventional score test in genetic association studies, one can directly generate the knockoff feature importance score per variant without the need to generate individual-level knockoffs for hundreds of thousands of samples. The method requires only summary statistics (i.e., Z-scores) from conventional GWAS while retaining the useful features of knockoff-based inference. The method additionally allows meta-analysis of studies with arbitrary sample overlap. We demonstrate its performance in empirical simulations and two applications: (1) meta-analysis study of Alzheimer's disease (AD) aggregating five genome-wide association studies, three whole-exome sequencing studies and one whole-genome sequencing study; (2) analysis of 1403 individual binary phenotypes from UK Biobank data on 408,961 samples with European ancestry. These analyses demonstrate the appealing properties of the proposed method for robust discovery of

additional loci and ability to localize putative causal variants at each locus. Additional discoveries made by the proposed method are further supported by functional enrichment analyses and single-cell transcriptomic analyses. The method is computationally efficient and required only 5.45 h on one central processing units (CPU) to analyze genome-wide summary statistics from the nine AD studies.

## Results
### Summary statistics-based multiple knockoff inference
We assume a study population of $n$ independent individuals and $p$ genetic variants. Let $\boldsymbol{X_i} = (X_{i1}, \ldots, X_{iq})$ be a vector of covariates, $\boldsymbol{G_i} = (G_{i1}, \ldots, G_{ip})$ be a vector of genotypes for the $i$th individual, and $Y_i$ be the phenotype with conditional mean $\mu_i$ given $\boldsymbol{X_i}$ and $\boldsymbol{G_i}$. A commonly used statistical model for modeling genetic association is the generalized linear model:

$$g(\mu_i) = \alpha_0 + \boldsymbol{\alpha}^T \boldsymbol{X_i} + \boldsymbol{\beta}^T \boldsymbol{G_i}, \qquad (1)$$

where $g(\mu) = \mu$ for a continuous trait, and $g(\mu) = \text{logit}(\mu)$ for a binary trait. Without loss of generality, we assume that both phenotype and genotype are centered and standardized to have mean 0 and variance 1. If there are covariates involved, $\boldsymbol{Y}$ can be centered at the conditional mean given the covariates. Conventional GWAS performs a marginal association test for each variant and controls for FWER. It tests against hypothesis $H_0 : Y_i \perp G_{ij}$ for $j = 1, \ldots, p$ via a score test. The per-sample score statistic can be written as $\boldsymbol{G_i}^T Y_i$. The Z-scores aggregating all samples can be written as

$$\boldsymbol{Z_{score}} = \frac{1}{\sqrt{n}} \boldsymbol{G}^T \boldsymbol{Y} \qquad (2)$$

where $\boldsymbol{G}$ is a $n \times p$ genotype matrix; $\boldsymbol{Y} = (Y_i, \ldots, Y_n)^T$. Typical knockoff-based inference contains four main steps: (1) generate one or multiple knockoffs per variant and per sample; (2) calculate the feature importance score for both original and knockoff variants, e.g. square of a Z-score, $\boldsymbol{Z_{score}^2}$ and the knockoff counterpart $\widetilde{\boldsymbol{Z}}_{score}^2$; (3) calculate the feature statistic by contrasting the feature importance scores for the original and knockoff variants; (4) implement the knockoff filter procedure to select significant variants with FDR control[7]. Although the Z-scores above are derived from a generalized linear model, it is worth noting that knockoff inference holds without the explicit model assumption. Unlike conventional score test or permutation analysis, knockoff-based inference performs conditional tests that account for linkage disequilibrium and controls FDR. It tests against hypothesis $H_0 : Y_i \perp G_{ij} | \boldsymbol{G_{i,-j}}$ for $j = 1, \ldots, p$, where $\boldsymbol{G_{i,-j}}$ includes all genetic variants except the $j$th variant.

The knockoff generation in step (1) can be computationally intensive. The main contribution of this paper is to show that for this particular form of feature importance score (e.g., the conventional score test as in genetic association studies), one can directly generate the knockoff feature importance score per variant without the need to generate individual-level knockoffs for hundreds of thousands of samples. Our method takes simple Z-scores as input and retains many useful features of knockoff-based inference, except for the flexibility to incorporate more sophisticated (non-linear) machine learning algorithms.

For a multiple-knockoff-based inference where each genetic variant is paired with $M$ knockoffs, we show that the knockoff counterpart for $\boldsymbol{Z_{score}}$ can be directly generated by

$$\widetilde{\boldsymbol{Z}}_{score} = \boldsymbol{P} \boldsymbol{Z_{score}} + \boldsymbol{E}, \text{ with } \boldsymbol{E} \sim \boldsymbol{N}(\boldsymbol{0}, \boldsymbol{V}), \qquad (3)$$

where $\widetilde{\boldsymbol{Z}}_{score} = (\widetilde{\boldsymbol{Z}}_{score}^m)_{1mM}$ is a $pM$ dimensional vector and each $\widetilde{\boldsymbol{Z}}_{score}^m$ is a $p$ dimensional vector of Z-scores corresponding to the $m$th group of knockoffs; $\boldsymbol{P}$ and $\boldsymbol{V}$ are $pM \times p$ and $pM \times pM$ matrices respectively obtained by solving a convex optimization problem (see "Methods"

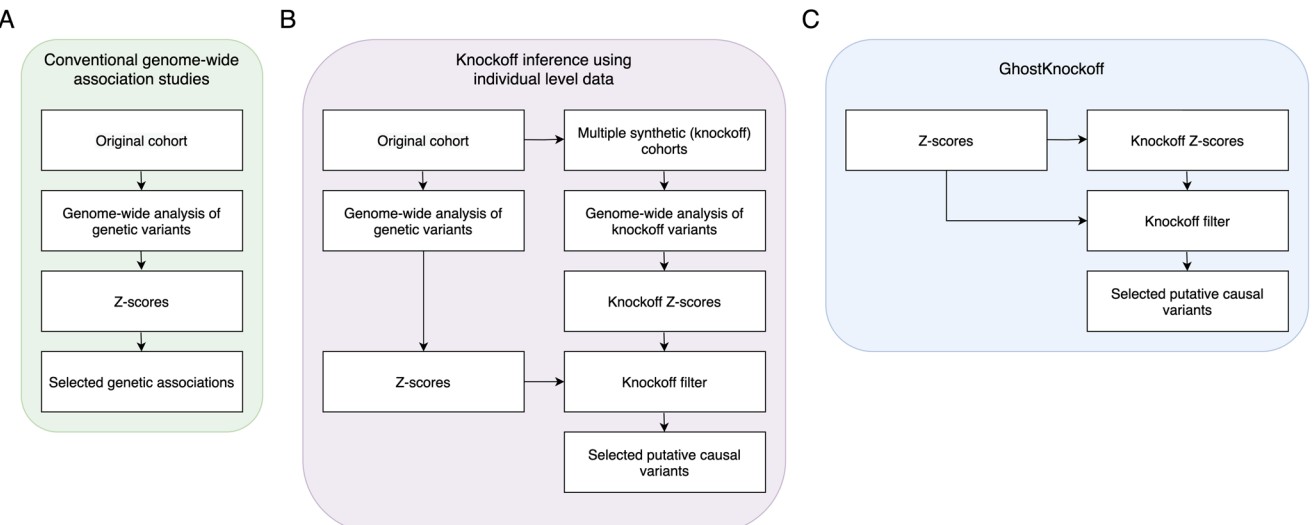

**Fig. 1 | Overview of GhostKnockoff.** We present the workflow of GhostKnockoff compared to conventional GWAS and knockoff inference based on same marginal test statistics using individual level data. **A** Conventional GWAS. **B** Knockoff inference using individual level data. We present the approach based on marginal test statistics. **C** The proposed GhostKnockoff using Z-scores from conventional GWAS as input.

section). Note that the matrices $\boldsymbol{P}$ and $\boldsymbol{V}$ are derived from the LD structure of the variants, which can be estimated by means of an external reference panel when individual-level data are not available. Intuitively, $\boldsymbol{P}$ can be viewed as a "projection matrix" that maps each Z-score from a marginal test to a Z-score that quantifies indirect effects through other variants due to linkage disequilibrium. Therefore, the contrast between $\boldsymbol{Z_{score}}$ and $\widetilde{\boldsymbol{Z}}_{\boldsymbol{score}}$ can prioritize causal variants that have a direct effect on the outcome of interest. This way, the Z-scores for knockoff variants can be efficiently obtained by "projecting" and sampling from a multivariate normal distribution. We show that the knockoff Z-scores generated by this approach are equivalent in distribution to those calculated based on individual-level knockoffs, thus enjoying all the desirable properties thereof. We present the details of knockoff filter in "Methods" and the workflow in Fig. 1.

GhostKnockoff was derived for a particular form of Z-score described above. In practice, we may obtain p-values from different statistical models (e.g. linear model, logistic model, mixed model etc., with different covariate adjustments) and from different tests (e.g. Wald's test, likelihood ratio test, score test etc.). In such cases, we can apply GhostKnockoff to Z-scores obtained by an inverse normal transformation of p-values multiplied by the direction of effect, i.e., $\text{sign} \times \Phi^{-1}(\text{pvalue}/2)$. In simulations, we have observed that the FDR control remained robust as long as the p-values were computed using tests with valid type I error rate. For example, for studies with related samples, the p-values can be computed using a mixed model that accounts for sample relatedness, such as GMMAT[14], SAIGE[15] and fastGWA-GLMM[16]. We further discuss the robustness of GhostKnockoff in the Discussion section.

**Meta-analysis of possibly overlapping studies**

Suppose Z-scores from $K$ independent studies with sample sizes $n_1, \ldots, n_K$ are available. We denote them as $\boldsymbol{Z_{1,score}}, \ldots, \boldsymbol{Z_{K,score}}$; $N = n_1 + \ldots + n_K$ is the total number of samples including possible duplicates. In general, the meta-analysis Z-score can be written as a weighted sum of individual study Z-scores

$$\boldsymbol{Z_{score}} = \sum_k w_k \boldsymbol{Z_{k,score}} \tag{4}$$

where $w_k$ is the weight assigned to the $k$th study[17,18]. The choice $w_k = \sqrt{n_k/N}$ corresponds to a conventional meta-analysis Z-score

weighted by sample size. Studies with overlapping samples are common in meta-analyses of genetic data, therefore we consider a weighting scheme that accounts for possible sample overlap and that maximizes the power of the meta-analysis. Intuitively, the optimal weights will up-weight those studies with higher independent contribution and down-weight the studies that largely overlap with others. In the "Methods" section, we show that the optimal weights $w_k$ are given by solving

$$minimize \sum_{1 \le i,j \le K} w_i w_j cor.S_{ij}, subject\ to \sum_k w_k \sqrt{n_k} = 1, w_k \ge 0, \tag{5}$$

where $cor.S_{ij}$ quantifies the study correlations due to sample overlap. We propose a method based on the knockoff framework to estimate $cor.S_{ij}$ (see "Methods" section). Note that, for case-control studies, $n_k$ can be replaced by $4/(\frac{1}{n_{k,case}} + \frac{1}{n_{k,control}})$ to better account for case-control imbalance[17].

Our proposal for meta-analysis of possibly overlapping studies via knockoffs is to mimic a pooled mega-analysis, by revising the knockoff generation to account for the possibility of duplicated samples. We first consider a scenario where all studies in the meta-analysis are homogeneous and have a shared LD structure. In the meta-analysis setting we compute the knockoff Z-scores as

$$\widetilde{\boldsymbol{Z}}_{\boldsymbol{score}} = \sum_k w_k * (\boldsymbol{PZ_{k,score}} + \gamma \boldsymbol{E_k}), \boldsymbol{E_k} \sim \boldsymbol{N}(\boldsymbol{0}, \boldsymbol{V}) \text{independently for all } k \tag{6}$$

where

$$\gamma = \sqrt{1 + \frac{N}{\hat{N}_{eff}} - \frac{\hat{N}_{eff}}{N}}, \frac{\hat{N}_{eff}}{N} = \frac{\sum_k w_k^2}{\sum_{1 \le i,j \le K} w_i w_j cor.S_{ij}} \tag{7}$$

$cor.S_{ij}$ quantifies the correlation between studies $i$ and $j$; $\{w_k\}_{1 \le k \le K}$ are the solution of the above quadratic optimization problem. Note that $\gamma \ge 1$ can be thought of as a "dependency factor" that accounts for sample overlap; meta-analysis of independent studies corresponds to $\gamma = 1$, when $cor.S_{ij} = 0$, $i \ne j$ and subsequently $\hat{N}_{eff} = N$. We present the details in the "Methods" section.

Next, we consider a more general scenario where there are $L$ groups of studies with different LD structures across groups (e.g. each

group could be of a different ancestry). We assume that each group $l$ includes $K_l$ (homogenous) possibly overlapping studies with sample sizes $n_{lk}$ and Z-scores $Z_{lk,score}$; $n_l = \sum_k n_{lk}$. In the "Methods" section, we show that the overall Z-score and its knockoff counterpart can be computed as

$$Z_{score} = \frac{1}{\sqrt{N}} \sum_{1 \le l \le L} \sqrt{n_l} * Z_{l,score}, \widetilde{Z}_{score} = \frac{1}{\sqrt{N}} \sum_{1 \le l \le L} \sqrt{n_l} * \widetilde{Z}_{l,score} \quad (8)$$

where for the $l$th group $Z_{l,score} = \sum w_{lk} Z_{lk,score}$; $\widetilde{Z}_{l,score} = \sum w_{lk} * (P_l Z_{lk,score} + \gamma_l E_{lk})$ with $E_{lk} \sim N(0, V_l)$ independently for all $l$ and $k$; $P_l$ and $V_l$ can be obtained by solving the same convex optimization problem in "Methods" section, using the LD structure of the corresponding group. Intuitively, we perform the knockoff generation for each group separately, and then aggregate the group Z-scores to compute an overall Z-score and its knockoff.

## Computational efficiency

GhostKnockoff is computationally efficient. Empirically, it only required 11.5 hours on average with one CPU to analyze a phenotype from the UK Biobank, and 5.45 hours to meta-analyze the nine AD genetic studies. This is significantly faster than the existing knockoff methods that require individual level data, which can take several days as reported in Sesia et al.[11] and He et al.[12]. Note that both the HMM method and the SCIT method for individual level knockoff generation has a model complexity $O(np)$. By comparison, directly generating knockoff Z-scores as in GhostKnockoff has a model complexity $O(p)$. A primary gain of computational efficiency is from using a reference panel to pre-compute required matrices ($P$ and $V$ as described above) for generating knockoff Z-scores; $P$ and $V$ are pre-computed using the correlation (LD) structure estimated from a reference panel. Moreover, the random Gaussian term $E$ can be pre-sampled given $V$. Therefore, the generation of knockoff Z-scores for a new study only involves few steps of simple matrix multiplications. For scenarios where we have Z-scores from multiple phenotypes or from multiple studies that share the same LD structure, the same pre-calculated matrices can be simultaneously applied. For new studies where the Z-scores are not readily available, there are many new advances in computing variant-level Z-scores and $p$-values efficiently for biobank scale data. The proposed method can leverage other analytical tools that efficiently compute variant-level Z-scores.

## Power and FDR simulations

We performed simulations to empirically evaluate the performance of GhostKnockoff, which include: (1) comparing knockoff inference based on summary statistics vs. knockoff inference based on individual level data; (2) evaluating the proposed method that accounts for sample overlap; (3) comparing multiple-knockoff inference vs. single knockoff inference; (4) comparing knockoff inference vs. conventional marginal association tests in terms of the prioritization of causal variants. Note that the power comparisons between knockoff FDR control and usual FWER control in a genome-wide setting have been extensively studied by Sesia et al. (2020) and He et al. (2021), and therefore we did not focus on these existing comparisons in this paper. Instead, our simulation study focuses on method comparison in a local region. We simulated genetic data directly using whole-genome sequencing (WGS) data from the Alzheimer's Disease Sequencing Project (ADSP). The ADSP WGS data (NG00067.v5) are jointly called by the ADSP consortium following the SNP/Indel Variant Calling Pipeline and data management tool (VCPA)[19]. We restricted the sampling to individuals with >80% European ancestry (estimated by SNPWeights v2.1 using reference populations from the 1000 Genomes Consortium[20,21]). For each replicate, we randomly drew individuals for two overlapping studies with 2500 individuals per study and 2000 genetic variants

randomly selected from a 1Mb region near the *APOE* region (chr19:44,909,011-45,912,650; hg38). We then restricted the simulation studies to variants with minor allele counts > 25 and to variants that are not tightly linked. We considered three levels of sample overlap: 0% (independent), 25% (moderate) and 50% (high), with 0%, 25% and 50% samples in each study being present in the other study. Details on the simulation studies can be found in the "Methods" section.

First, we compare GhostKnockoff (GhostKnockoff-S and Ghost-Knockoff-M, where S or M represents single and multiple knockoff (five knockoff copies per variant), respectively) with the second-order knockoff generator proposed by Candès et al. that requires individual-level data (IndividualData Knockoff-S). We also extend the second-order knockoff generator to the multiple knockoff setting and include it in the comparison (IndividualData Knockoff-M). The methods that require individual level data were applied to an oracle pooled dataset that only contains unique samples. For a fair comparison, all methods are based on the same feature importance scores, i.e. the squared Z-scores from a marginal score test for association. We present the results in Fig. 2 (A–F). As shown, all methods have valid FDR control. In terms of power, the proposed methods based on summary statistics (GhostKnockoff-M/S) have consistent power as methods that require individual level data (IndividualData Knockoff-M/S). The results also hold when there is 25%/50% sample overlap. When there is sample overlap, the power of GhostKnockoff-M/S becomes slightly lower than IndividualData Knockoff-M/S. A likely explanation is that the study correlation in this simulation setting is over-estimated and, subsequently, the proposed method becomes slightly conservative. In real data applications, we will use genome-wide Z-scores to estimate the sample correlation, which should be more accurate. This simulation study also confirms the higher power of multiple knockoff inference (GhostKnockoff-M) relative to single knockoff inference (Ghost-Knockoff-S) at low target FDR (e.g. 0.05/0.10) and in the scenario with small number of causal variants (10 in our setting) as shown by Gimenez and Zou (2018)[22]. This is because the detection threshold of the knockoff filter (the necessary number of independent signals $\approx \frac{1}{M \times \text{target FDR}}$, where $M$ is the number of knockoff copies per variant) is lower for multiple knockoffs compared to single knockoff. The power of multiple knockoff inference will be eventually comparable to single knockoff inference at high target FDR or in a genome-wide analyses of polygenic traits as shown in He et al.[12].

Second, we compare GhostKnockoff-M (target FDR = 0.1) with conventional marginal tests used in GWAS, adjusted by Bonferroni correction for FWER control (IndividualData marginal test-Bonferroni; target FWER = 0.05) or Benjamini-Hochberg procedure for FDR control (IndividualData marginal test- Benjamini-Hochberg; target FDR = 0.1). We show results for the setting with 25% sample overlap. Since the FDR control is more liberal than the FWER control, we focus on the prioritization of causal variants in this simulation study. For each replicate, we calculate the proportion of identified variants being causal (1-FDR) for each method. We observed that the causal proportion for GhostKnockoff-M is substantially higher than for conventional marginal tests, because GhostKnockoff-M performs a conditional test, and therefore can properly account for LD (Fig. 2G, H). Furthermore, the causal proportion using the Benjamini–Hochberg procedure is lower than that using the Bonferroni correction. Since the Benjamini-Hochberg procedure assumes positive dependence among tests, it may fail to control FDR under more complex LD structure, which might result in increased false positive rates. Therefore the conventional marginal testing with Benjamini-Hochberg adjustment does not provide a valid approach for GWAS.

Finally, we compare multiple knockoff inference (GhostKnockoff-M) with single knockoff inference (GhostKnockoff-S) in terms of the randomness due to sampling knockoff copies, referred to as stability of knockoff inference. We show results for the setting with 25% sample

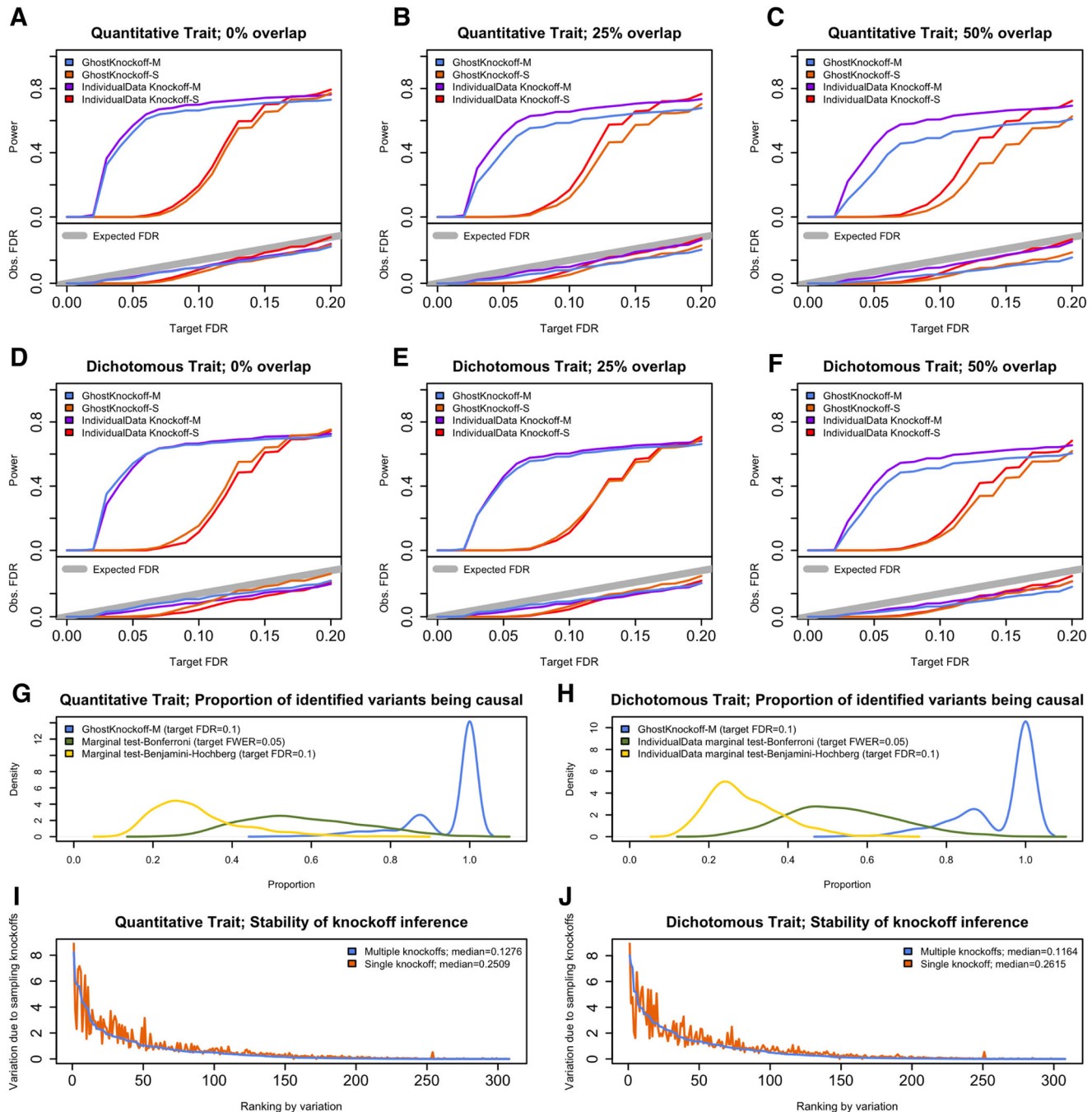

**Fig. 2 | Empirical simulation studies for power, FDR and stability.** Two cohorts are randomly sampled from the same population. **A–F.** Power and FDR based on 1000 replicates for different types of traits (quantitative and dichotomous) and different levels of sample overlap (0%/25%/50%), with different target FDR varying from 0 to 0.2. GhostKnockoff-M/S: the proposed multiple/single knockoff method based on the meta-analysis of Z-scores calculated separately from each individual cohort. IndividualData Knockoff-M/S: knockoff inference based on individual level data. **G, H**. Prioritization of causal variants. **I, J**. Stability of knockoff inference, with 25% overlap and 20% unobserved variants per study. The stability is quantified as the standard deviation of feature statistics across 1000 replicates due to randomly sampling knockoffs for a given dataset.

overlap. We fixed the genotype data and phenotype data, and repeatedly performed knockoff inference. For each replicate, we computed the W-statistic for each variant. Then we calculated the standard deviation of the W-statistics per variant to quantify the variation due to sampling knockoffs. The multiple knockoff procedure shows less randomness due to sampling knockoffs compared to single knockoff inference (Fig. 2I–J).

We have also performed additional comparisons with other existing knockoff generators that require individual level data, including the knockoff generator for Hidden Markov Models

(HMMs) proposed by Sesia et al. (2019) with number of states S = 50, and the sequential knockoff generator proposed by He et al. (2021). We observed similar results as in comparison with the second-order knockoff generator above (Supplementary Fig. 1). In Supplementary Fig. 2, we present an empirical evaluation of the robustness to study-specific rare variants, where 10%/20% rare variants (minor allele frequency < 0.01) are randomly set to be unobserved in each cohort. We observed that the method remains valid, but requires slight modifications as described in the "Methods" section.

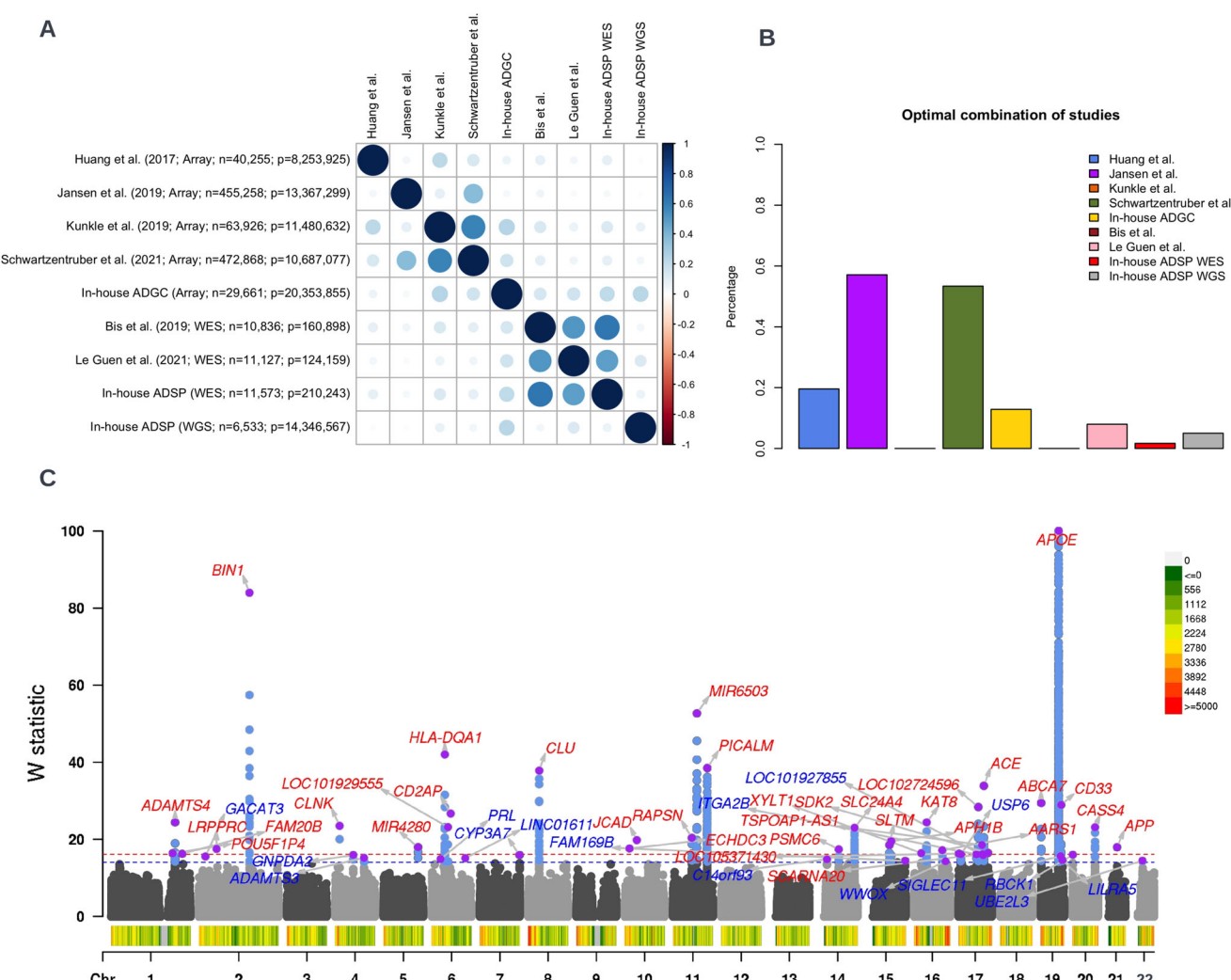

**Fig. 3 | Meta-analysis of Alzheimer's disease studies. A** Study correlations estimated using the proposed method. For each study, we present sequencing technology, sample size and number of variants. **B** Optimal combination of studies estimated using the proposed method. Each bar presents the weight per study in percentage, i.e. weight per study divided by the summation of all weights. **C** We present the Manhattan plot of W statistics (truncated at 100 for clear visualization) from GhostKnockoff with target FDR at 0.05 (red) and 0.10 (blue). The results are based on the optimal weights combining the nine studies. Variant density is shown at the bottom of Manhattan plot (number of variants per 1Mb).

## Meta-analysis of Alzheimer's disease genetics

We applied GhostKnockoff to aggregate summary statistics from nine overlapping large-scale array-based genome-wide association studies, and whole-exome/-genome sequencing studies. Specifically, the studies include (1) The genome-wide survival association study performed on 14,406 AD case samples and 25,849 control samples by Huang et al.[23]; (2) The genome-wide meta-analysis of clinically diagnosed AD and AD-by-proxy (71,880 cases, 383,378 controls) by Jansen et al.[24]; (3) The genome-wide meta-analysis of clinically diagnosed AD (21,982 cases, 41,944 controls) by Kunkle et al.[25]; (4) The genome-wide meta-analysis by Schwartzentruber et al.[26], aggregating Kunkle et al. 2019 and UK Biobank based on a proxy AD phenotype; (5) In-house genome-wide associations study of 15,209 cases and 14,452 controls aggregating 27 cohorts across 39 SNP array datasets, imputed using the TOPMed reference panels[27]; (6, 7) Two whole-exome sequencing analyses of data from The Alzheimer's Disease Sequencing Project (ADSP) by Bis et al.[28] (5740 cases, 5096 controls), and Le Guen et al.[29] (6008 cases, 5119 controls); (8) In-house whole-exome sequencing analysis of ADSP (6155 cases, 5418 controls); (9) In-house whole-genome sequencing analysis of the 2021 ADSP release[30] (3584 cases, 2949 controls). All studies focused on individuals with European ancestry. We used LD matrices estimated using the high coverage whole-

genome sequencing data of the expanded 1000 Genomes Project[31]. Due to the relatively small sample size of the 1000 Genomes Project (503 individuals of European ancestry) used to estimate the LD matrices, we restrict the analyses to common and low-frequency variants with minor allele frequency >1%.

We present the estimated study correlations $cor.S_{ij}$ in Fig. 3A, and the estimated optimal weights in Fig. 3B. The correlation results are consistent with our knowledge of overlap and other factors, such as differences in phenotype definition, analysis strategies (e.g. statistical model), and quality control, that can affect the correlations between these studies (see more details in Supplementary Materials). Similarly, the weighting scheme up-weighted studies that are large in size and carry independent information, and down-weighted studies that largely overlap with others. In the Supplementary Materials, we discuss the consistency between the estimated study correlations and similarities in the design of these studies.

We present the results of the meta-analyses of the nine studies in Fig. 3C. We define two loci as independent if they are at least 1Mb away from each other. We adopt the most proximal gene's name as the locus name, recognizing that it is not necessarily the most likely causal gene. Our analysis identified in total 34 loci significant at FDR 0.05 and 50 loci significant at FDR 0.1. Supplementary Table 2 summarizes the 34

loci at FDR 0.05. The results show that most of the associations exhibit suggestive signals in individual studies, and most identified loci, including existing and new ones, have the same direction of effects across all studies, except very few loci where one dataset has an opposite direction of effect, although not significant, relative to other studies. Several new genes that were not reported in the latest AD meta-analyses including Jansen et al. (2019), Kunkle et al. (2019) and Schwartzentruber et al. (2021) are worth mentioning. For example, *LRPPRC* (leucine-rich pentatricopeptide repeat motif containing protein) and *APP* (Amyloid beta precursor protein) have support for their possible involvement in AD from multiple studies[32]. Furthermore, Hosp et al. (2015) identified *LRPPRC* as a preferential interactor of *APP* carrying the so-called Swedish mutation (APPsw), which causes early-onset AD[33]. Among the new genes, *TREM1, CYP3A7, SIGLEC11, IL34, RBCK1, C16orf92, WWOX* are within 1MB of novel loci reported in recent studies by Wightman et. al. (2021)[34] and Bellenguez et al.[35].

To validate the results from the knockoff inference, we adopted an alternative strategy when an independent replication study is not available. Specifically, we applied the method to a subset of samples and show that the identified variants are also replicated when we increase the sample size. We considered GhostKnockoff analysis of Kunkle et al. (2019), Schwartzentruber et al. (2021), and all nine studies. Note that data from Kunkle et al. (2019) is a subset of Schwartzentruber et al. (2021); Schwartzentruber et al. (2021) is a subset of the meta-analysis. We considered the replication of genetic variants. A genetic variant is replicated if the same variant is also identified in the larger study with the same direction of effect and a smaller *p*-value. We present the results for FDR = 0.05 and FDR = 0.10 in Supplementary Table 1 and Supplementary Fig. 3.

At FDR=0.05, we observed that 338 out of 385 (87.8%) variants by GhostKnockoff analysis of Kunkle et al. (2019) are also replicated in the analysis of Schwartzentruber et al. (2021); 447 out of 634 (70.5%) variants identified by GhostKnockoff analysis of Schwartzentruber et al. (2021) are also replicated in the proposed meta-analysis of all nine studies. At FDR = 0.10, 370 out of 448 (82.6%) variants identified by GhostKnockoff analysis of Kunkle et al. (2019) are also replicated in the analysis of Schwartzentruber et al. (2021); 510 out of 724 (70.4%) variants identified by GhostKnockoff analysis of Schwartzentruber et al. (2021) are also replicated in the proposed meta-analysis of all nine studies. Overall, we conclude that the proposed GhostKnockoff (conditional test + FDR control), though systematically different from the conventional GWAS, is a valid approach to make reproducible genetic discoveries.

Finally, we present the results based on sample size weighted combination as opposed to the proposed optimal weights in Supplementary Fig. 4. We observed that GhostKnockoff with standard weights identified 29 loci at FDR 0.05 and 46 loci at FDR 0.10, while GhostKnockoff with proposed weighting scheme identified 34 loci at FDR 0.05 and 50 loci at FDR 0.10.

### Single-cell transcriptomics differential expression analyses validate proximal AD genes

For the proximal genes (nearest genes to the lead variant) corresponding to the 50 loci identified at FDR 0.10 in Fig. 3, we performed differentially expressed gene (DEG) analyses using single-cell RNA sequencing data (scRNAseq) from 143,793 single-nucleus transcriptomes from 17 hippocampus (8 controls and 9 AD cases) and 8 cortex samples (4 controls and 4 AD cases)[36]. We performed the DEG analysis stratified by 14 cell types, spanning major brain cell types (e.g., neurons, astrocytes, microglia) and cell types that reside in the vascular, perivascular, and meningeal compartments, including endothelial cells, pericytes and smooth muscle cells, fibroblasts, perivascular macrophages and T cells. We adjusted for age, batch, cellular detection rate, and for within-sample correlation by including sample dummy variables as covariates. We used this fixed effect model instead of a random effect model because the number of clusters is

small relative to the total number of cells. Among the 50 proximal genes, 38 had expression measurements in the scRNAseq dataset. We considered *p*-value threshold 0.05 for suggestive signals and a more stringent Bonferroni correction 0.05/38 = 0.0013 for significant signals (more details on the analyses are available in the "Methods" section).

We show the scRNAseq results (−log10(*p*-value) vs. log2 fold change) in Fig. 4A. Overall, we observed a consistently higher proportion of differentially expressed genes for the proximal genes compared to all other 23496 genes that are observed in the scRNAseq data (background genes), across the 14 cell types (Fig. 4B). Specifically, we found that 25/38 (65.79%) genes exhibit suggestive signal ($p < 0.05$) in at least one cell type, a significantly higher proportion compared with the background genes (41.73%; $p = 4.4 \times 10^{-3}$ by Fisher's exact test; Fig. 4C). Among the genes identified by GhostKnockoff at FDR = 0.05, 69.23% exhibit suggestive signals ($p < 0.05$), similar to the proportion for the proximal genes identified at FDR = 0.10. We also observed that the DEG signals are more pronounced for the genes identified by GhostKnockoff compared to background genes (Fig. 4D; *p*-values are generally smaller), showing a strong enrichment of DEG signals for the proximal genes that reside in the loci identified by GhostKnockoff.

### Phenome-wide analysis of UK Biobank data

We applied GhostKnockoff separately to each of 1403 binary phenotypes from the UK Biobank data with 408,961 white British participants (European ancestry). In this analysis, GhostKnockoff reduces to a knockoff inference based on summary statistics from a single study. We collected existing Z-scores calculated by Zhou et al. (2018) using SAIGE, a method that controls for case-control imbalance and sample relatedness[15]. GhostKnockoff was applied to each phenotype separately to select associated genetic variants at FDR 0.1. Similar to the AD genetics analysis, we restrict the analyses to common and low-frequency variants with minor allele frequency >1%.

We aim to illustrate two properties of GhostKnockoff inference that are systematically different from a conventional GWAS analysis. First, GhostKnockoff controls for FDR while conventional GWAS controls FWER; Second, GhostKnockoff performs a conditional test while conventional GWAS performs a marginal association test. We expect that the more liberal FDR control will lead to more associated loci, but the conditional test will reduce false positive findings due to LD confounding at each locus. For each phenotype, we count the number of independent associated loci, i.e. loci more than 1Mb away from each other. Within each locus, we count the number of genetic variants passing the FDR 0.1 threshold. We compare GhostKnockoff results and results from conventional GWAS using SAIGE (with *p*-value threshold $5 \times 10^{-8}$) in terms of the number of identified independent loci (FDR vs. FWER) and the number of identified variants per locus (conditional test vs. marginal test) (Fig. 5).

GhostKnockoff identifies generally more loci per disease phenotype (4.18-fold more discoveries on average, Figs. 5A, C) relative to GWAS. This is not surprising given that we identify significant variants at a more liberal threshold of FDR 0.1. More interesting, GhostKnockoff identifies less genetic variants within each locus (52% less variants on average after accounting for LD, Fig. 5B, C), even though FDR control is more liberal than the FWER control. We additionally present the results stratified by phecode category (phecodes are grouped into different categories as in SAIGE) in Fig. 5D, E[15]. Again, we observed that the proposed method consistently exhibits more loci and less genetic variants within each locus across disease categories.

To evaluate the functional effect of the identified variants, we performed functional enrichment analysis using 19 functional scores included in regBase[37], including: CADD[38], DANN[39], FATHMM-MKL[40], FunSeq2[41], Eigen[42], Eigen_PC[42], GenoCanyon[43], FIRE[44], ReMM[45], LINSIGHT[46], fitCons[47], FATHMM-XF[48], CScape[49], CDTS[50], DVAR[51], FitCons2[52], ncER[53], Orion[54] and PAFA[55]. All scores are on the Phred

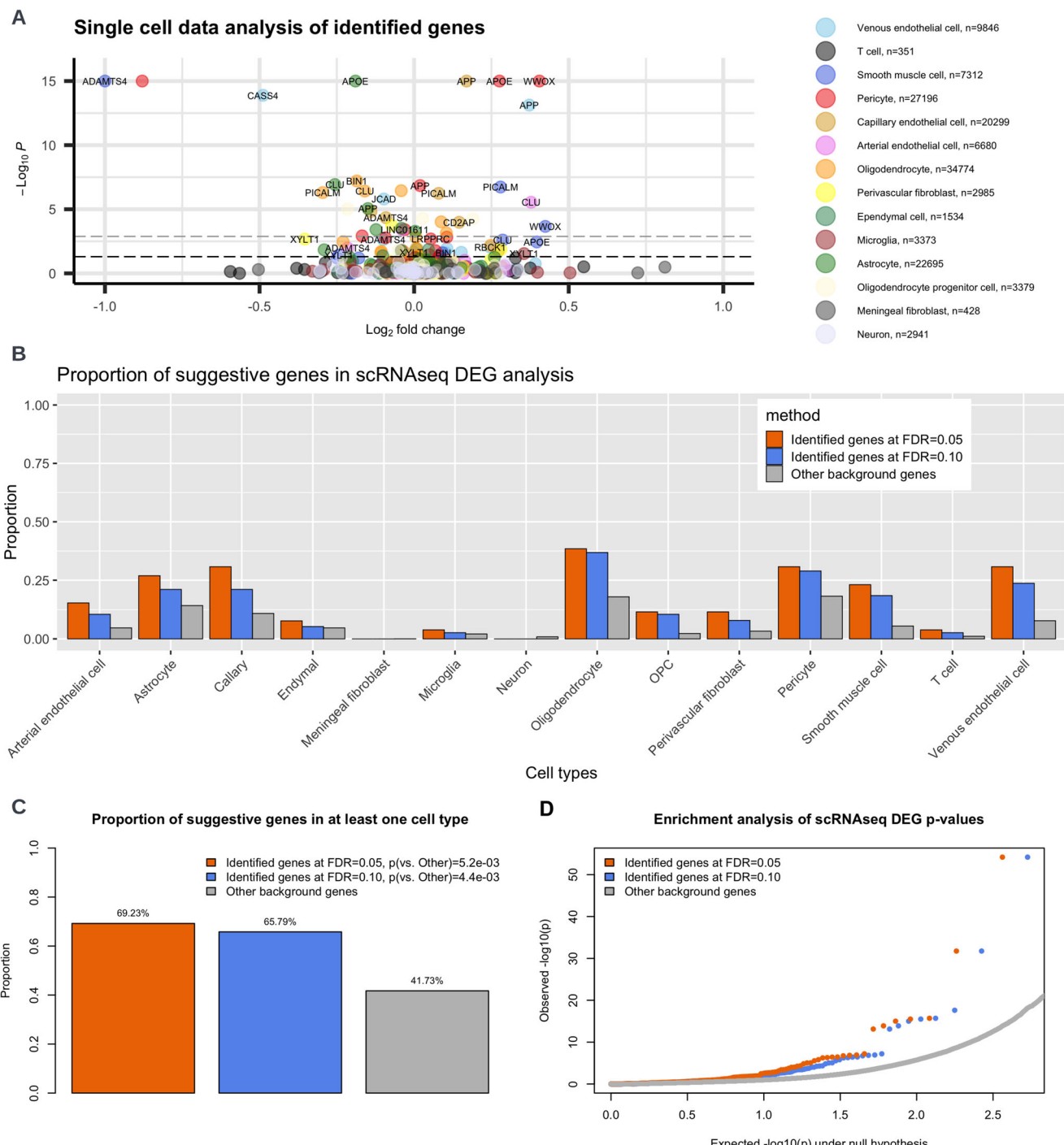

**Fig. 4 | Single-cell RNAseq data (n = 143793) analysis of the identified proximal AD genes. A** Differentially expressed genes (DEG) analysis using MAST implemented in Seurat, comparing Alzheimer's disease cases (AD) with healthy controls. Each dot represents a gene. Colors represent different cell types. OPC: Oligodendrocyte progenitor cell. The black dashed line corresponds to p-value cutoff 0.05; the gray dashed line corresponds to p-value cutoff 0.05/38 (number of candidate genes) which accounts for multiple comparisons. For visualization purposes,

−log10(p) values are capped at 15 and abs(log2(fold change)) values are capped at 1.0. Positive log2 fold change corresponds to higher expression level in AD. **B** Proportion of suggestive genes stratified by cell types. **C** Proportion of suggestive genes in at least one cell type. P-values are calculated with two-sided Fisher's exact test. **D** Enrichment analysis of DEG nominal p-values relative to background genes. P-values are calculated by MAST implemented in Seurat.

scale. We partitioned the identified variants into three sets: 1. Variants identified by conventional GWAS only; 2. GWAS discoveries that overlap with knockoff inference; 3. Additional discoveries made by knockoff inference. Each identified variant was MAF matched with 10 randomly selected background variants on the same chromosome. For each major disease category, we calculated the ratio between the average functional score of variants in a set and the average functional

score of background variants. A ratio higher than one indicates enriched functional effects of the identified variants. We present the results in Fig. 5F.

We observed that the additional discoveries made by knockoff inference are significantly enriched in higher functional scores of FunSeq2, Eigen_PC, GenoCanyon, FIRE, ReMM, LINSIGHT, fitCons, DVAR, fitCons2, and PAFA (p-values with one-sample t-test are shown

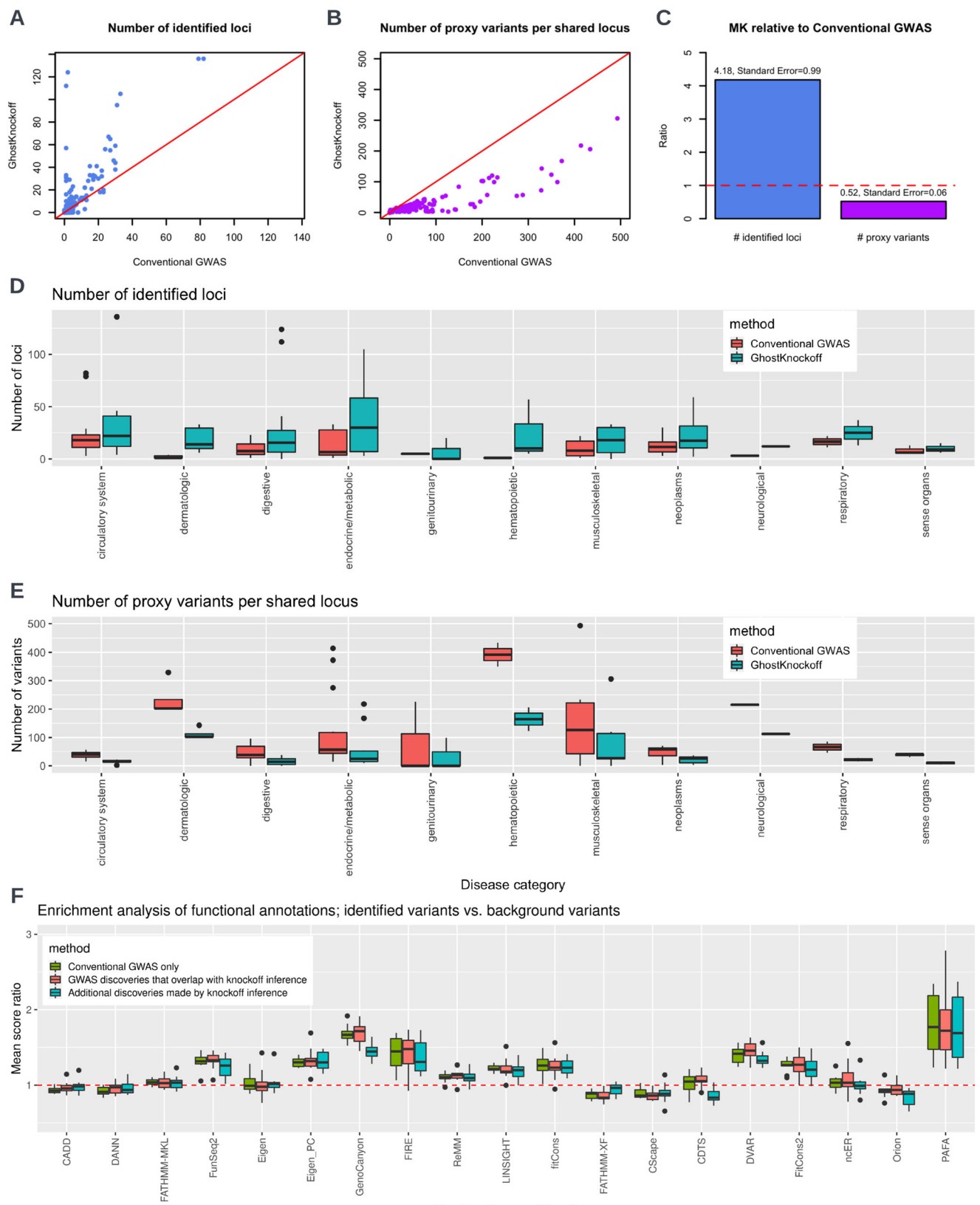

in Supplementary Fig. 5). Note that most of these functional annotation scores were proposed to predict regulatory effects of non-coding variants. This is expected given that most variants identified by GWAS and knockoff inference are in non-coding regions. Interestingly, PAFA, which prioritizes non-coding variants associated with complex diseases, shows the highest enrichment. These results illustrate that the additional discoveries made by knockoff inference, though weaker in terms of effect sizes, have putative regulatory effects on traits.

## Discussion

We have proposed GhostKnockoff to perform knockoff-based inference without generating any individual-level knockoff variants.

**Fig. 5 | Phenome-wide Analysis of 1403 binary phenotypes from UK biobank data with 408,961 white British participants with European ancestry.**
**A**, **B** Comparison between conventional GWAS and GhostKnockoff. **C** Summary of (**A**) and (**B**). For each phenotype, we calculated the ratio between the total number of identified loci/ the average number of proxy variants per shared locus by GhostKnockoff and by conventional GWAS (capped at 500 for better visualization). Panel (**C**) presents the average ratio (as in (**A**) and (**B**)) across 1403 phenotypes. The standard error is calculated as $\frac{\text{standard deviation of the ratio}}{\sqrt{\text{total number of phenotypes}-1}}$. **D** Distribution of the number of identified loci. We present boxplot (median and 25%/75% quantiles) for each disease category. **E** For loci identified by both conventional GWAS and the proposed method, we present median and 25%/75% quantiles of the number of identified variants per locus. For visualization purposes, we present the results for disease phenotypes with ≥5 loci identified by either conventional GWAS or the proposed multiple knockoff inference for panels (**D**, **E**). **F** Functional score of variants identified by GhostKnockoff compared to that of genome-wide background variants. Each data point in the boxplot corresponds to the average score of one disease category. The boxplot presents median and 25%/75% quantiles.

GhostKnockoff can be applied to commonly available summary statistics from conventional GWAS to improve the power to identify additional, potentially weaker, associations and to prioritize the causal variants at each locus. Additionally, GhostKnockoff can be applied to meta-analysis of possibly overlapping studies. In applications to phenome-wide analyses of UK Biobank data and a meta-analysis of Alzheimer's disease studies, we identified loci that were missed by conventional marginal association tests with improved precision. The additional discoveries are supported by functional enrichment analyses and single-cell transcriptomic analyses. These results demonstrate the improved performance of GhostKnockoff in distinguishing small effect loci that are potentially functional from noisy background genome.

As with fine-mapping studies, using in-sample LD information is best[56]. Oftentimes with large meta-analyses in-sample LD is unavailable, then the LD structure can be estimated from an external reference panel. Similar to other summary statistics-based methods, the current method assumes a matched reference panel to ensure the equivalence between knockoff inference based on individual level data and GhostKnockoff. Mismatch between the LD in the target cohort and the reference panel can increase the FDR. In our empirical studies and real data analyses, we found that the LD structure estimated from the appropriate population in the 1000 Genome data is a reasonably good approximation for common and low-frequency variants (MAF ≥1%). The effect of LD misspecification is local, and therefore may affect more the ability to prioritize the causal variants at each locus, and less so the genome-wide locus discovery. For lower frequency variants, larger reference panels, such as the TOPMed, gnomAD and Pan UKBB, have become increasingly available. Recent studies have shown that imputation quality of rare variants can be significantly improved by using these reference panels. We expect that the performance of GhostKnockoff on lower frequency variants can be improved similarly by leveraging these reference panels to better estimate the LD structure[57–59].

Another limitation of the current method is the practical aspect on how to deal with highly correlated variants. Although the knockoff method helps to prioritize causal variants over associations due to LD, it is difficult or impossible to distinguish causal genetic variants from highly correlated variants, e.g. the variants in regions of high LD such as the major histocompatibility complex (MHC). The presence of tightly linked variants can diminish the power to identify the causal ones. The current implementation applied a hierarchical clustering of genetic variants prior to the analysis and then randomly selected one variant in each cluster. Although this strategy ensures that each variant has a representative variant included in the analysis, the statistical power can be suboptimal when the underlying causal variant is not selected. Alternatively, the group knockoff filter which groups variants and thus requires exchangeability at the group rather than variant level can be used[60,61]. It would be of interest to incorporate group knockoffs into GhostKnockoff for improved power.

We focus on sample overlap because it is one of the main sources of study correlation, but the method can be more general to quantify study correlation due to other factors such as data generation, genotype coverage, imputation, and phenotype definition. In fact, the proposed estimation of study correlation is valid if the correlation of genome-wide Z-scores is correctly inferred. Our proposed method for estimating the correlation is data driven. If other factors increase/decrease the correlation, we think that the data-driven estimation will remain valid. However, we do require that i-values from different datasets are valid, and correctly reflecting the same disease-genetic association. If the p-values in the original study are deflated or inflated, the meta-analysis results can be biased subsequently. In addition, the dependency factor for knockoff generation was derived based on sample overlapping. In practice, we found that the analysis based on this dependency factor reasonably reflects other factors, but the theoretical guarantee will require future investigations.

We note that GhostKnockoff was derived for a particular form of Z-score, where both features and outcomes are standardized with mean zero and standard deviation one, and samples within each study are assumed independent. In practice available Z-scores can be based on different statistical models (e.g. linear model, logistic model, mixed model etc., with different covariate adjustments) and different tests (e.g. Wald's test, likelihood ratio test, score test etc.). Using empirical simulation studies, we observed that GhostKnockoff is robust to such variations (Fig. 2). Intuitively, the Z-scores from different analytical procedures share very similar joint distribution, with a similar marginal distribution and correlations mainly determined by LD. The proposed generation of knockoff Z-scores derived based on a particular form of Z-score $\frac{1}{\sqrt{n}} \boldsymbol{G}^T \boldsymbol{Y}$ provides a reasonable approximation and therefore the empirical FDR is under control. However, the theoretical justification of the robustness of GhostKnockoff remains unclear and it will be important to study in the future.

## Methods

### Knockoff-based inference using summary statistics

Our meta-analysis method is based on the second-order knockoff generator, which was initially developed for Gaussian distribution and then shown to remain empirically robust to certain deviations if the estimated second-order moments are sufficiently close to those of the underlying distribution.

The proposed knockoff-based inference using summary statistics attempts to mimic the inference based on individual-level data. For single-knockoff, we show in Supplementary Materials that the two methods are equivalent if we directly generate

$$\widetilde{\boldsymbol{Z}}_{score} = \boldsymbol{P}\boldsymbol{Z}_{score} + \boldsymbol{E}, \text{ with } \boldsymbol{E} \sim N(\boldsymbol{0}, \boldsymbol{V}) \tag{9}$$

$$\boldsymbol{P} = \left(\boldsymbol{I} - \boldsymbol{D}\boldsymbol{\Sigma}^{-1}\right), \boldsymbol{V} = 2\boldsymbol{D} - \boldsymbol{D}\boldsymbol{\Sigma}^{-1}\boldsymbol{D} \tag{10}$$

where $\boldsymbol{I}$ is a $p \times p$ identity matrix; $\boldsymbol{\Sigma}$ is the correlation matrix of $\boldsymbol{G}_i$ that characterizes the linkage disequilibrium; $\boldsymbol{D} = \text{diag}(s_1, \ldots, s_p)$ is a diagonal matrix obtained by solving the following convex optimization problem:

$$\text{minimize} \sum_{j=1}^{p} |1 - s_j|, \text{ subject to} \begin{cases} 2\boldsymbol{\Sigma} - \boldsymbol{D} \succcurlyeq 0, \\ s_j \geq 0, 1 \leq j \leq p \end{cases} \tag{11}$$

This way, we can directly simulate the knockoff Z-scores from a multivariate normal distribution. For a given dataset, the knockoff

Z-scores will follow the same distribution as those calculated based on generating individual-level knockoffs. Therefore, the summary statistics-based knockoff generation retains the properties of knockoff-based inference.

When $p$ is ultra-high-dimensional as in genetic studies with millions of variants, it is not feasible to operate with genome-wide correlation matrix $\boldsymbol{\Sigma}$. In practice, we divided the genome into blocks that can be loaded into memory and performed the intermediate calculation of the knockoff statistics. Then the knockoff statistics from all blocks are aggregated for a genome-wide feature selection. This practical solution does not model inter-block correlation. Consequently, the current method cannot attenuate the confounding effect of long-range LD.

### Extension to multiple knockoffs

Here we extend the single-knockoff approach to the case with multiple knockoffs where the original feature and multiple knockoffs are simultaneously exchangeable[12,62]. We show in Supplementary Materials that the multiple-knockoffs counterpart for a Z-score can be directly generated by

$$\widetilde{Z}_{score} = PZ_{score} + E, \text{ with } E \sim N(0, V) \tag{12}$$

$$P = \begin{pmatrix} I - D\Sigma^{-1} \\ \dots \\ I - D\Sigma^{-1} \end{pmatrix}, V = \begin{pmatrix} C & C-D & \dots & C-D \\ C-D & C & \dots & C-D \\ \dots & \dots & \dots & \dots \\ C-D & C-D & \dots & C \end{pmatrix} \tag{13}$$

where $\widetilde{Z}_{score}$ is a $pM$-dimensional vector; $I$ is a $p \times p$ identity matrix; $\boldsymbol{\Sigma}$ is the correlation matrix of $G_i$ that characterizes the linkage disequilibrium; $C = 2D - D\Sigma^{-1}D$; $D = \text{diag}(s_1, \dots, s_p)$ is a diagonal matrix given by solving the following convex optimization problem:

$$\text{minimize} \sum_j |1 - s_j|, \text{ subject to } \begin{cases} \frac{M+1}{M}\Sigma - D \succcurlyeq 0, \\ s_j \geq 0, 1 \leq j \leq p. \end{cases} \tag{14}$$

### Knockoff filter to define the threshold $\tau$ and Q-value for FDR control

After the knockoff Z-scores are generated, we calculate the feature importance score as the element-wise square of Z-scores,

$$T = Z_{score}^2, T^m = \left(\widetilde{Z}_{score}^m\right)^2 \tag{15}$$

and the knockoff statistics

$$\kappa_j = \underset{0 \leq m \leq M}{\arg\max} T_j^m, \tau_j = T_j^{(0)} - \underset{1 \leq m \leq M}{\text{median}} T_j^{(m)} \tag{16}$$

where $m$ indicates the $m$th knockoff. For the $j$th variant, $\kappa_j$ denote the index of the original (denoted as 0) or the knockoff feature that has the largest importance score; $\tau_j$ denotes the difference between the largest importance score and the median of the remaining importance scores; $T_j^{(m)}$ is corresponding to the order statistics with $T_j^{(0)} \geq \dots \geq T_j^{(m)}$. $\kappa$ and $\tau$ obey the a property similar to the "flip-sign" property in the single knockoff scenario[12,62]. In the multiple knockoff scenario, $\kappa_j$ plays a role as the sign, and $\tau_j$ quantifies the magnitude that is invariant to swapping. Subsequently, we define a $W$-statistic to quantify the magnitude of effect on the outcome as

$$W = \left(T - \underset{1 \leq m \leq M}{\text{median}} T^m\right) I_{T \geq \underset{1 \leq m \leq M}{\max} T^m} \tag{17}$$

Variants with $W > \tau$ are selected, where $\tau$ is the threshold calculated by the knockoff filter. We note that this $W$-statistic is different from the knockoff statistics in the original model-X knockoff paper[7]. Instead, it is a function of the exact knockoff statistics that obey the "flip sign" property in the context of multiple knockoff inference. We use it as a convenient and intuitive representation of the magnitude of association. We present the exact knockoff statistics that obey the "flip sign" property and the corresponding knockoff filter in the "Methods" section.

We define the threshold for the knockoff filter as

$$\tau = \min\left\{ t > 0 : \frac{\frac{1}{M} + \frac{1}{M}\#\{\kappa_j \geq 1, \tau_j \geq t\}}{\#\{\kappa_j = 0, \tau_j \geq t\}} \leq q \right\} \tag{18}$$

In addition, we define the Q-value for a variant with statistics $\kappa = 0$ and $\tau$ as

$$q = \min_{t \leq \tau} \frac{\frac{1}{M} + \frac{1}{M}\#\{\kappa_j \geq 1, \tau_j \geq t\}}{\#\{\kappa_j = 0, \tau_j \geq t\}} \tag{19}$$

where $\frac{\frac{1}{M} + \frac{1}{M}\#\{\kappa_j \geq 1, \tau_j \geq t\}}{\#\{\kappa_j = 0, \tau_j \geq t\}}$ is an estimate of the proportion of false discoveries if we are to select all variants with feature statistic $\kappa_j = 0, \tau_j \geq t$, which is the knockoff estimate of FDR. For variants with $\kappa \neq 0$, we define $q = 1$ and they will never be selected. Selecting variants with $W > \tau$ where $\tau$ is calculated at target FDR$=\alpha$ is equivalent to selecting variants with $q \leq \alpha$.

### Meta-analysis of independent studies via knockoffs

Suppose Z-scores from $K$ independent studies with sample sizes $n_1, \dots, n_K$ are available, denoted as $Z_{1,score}, \dots, Z_{K,score}$. We define the meta-analysis Z-score as

$$Z_{score} = \frac{1}{\sqrt{N}} \sum_k \sqrt{n_k} * Z_{k,score} = \frac{1}{\sqrt{N}} \sum_k S_k \tag{20}$$

where $N = \sum_k n_k$ is the total number of samples; $S_k$ is the score test statistic for the $k$-th study. Note that the meta-analysis Z-score is also a summation of sample score statistics, where the correlation structures across different studies are assumed to be the same. Following the same derivation for a single study, we can generate the knockoff feature importance by

$$\begin{aligned} \widetilde{Z}_{score} &= \frac{1}{\sqrt{N}} \sum_k \sqrt{n_k} * (PZ_{k,score} + E_k) \\ &= \frac{1}{\sqrt{N}} P \sum_k \sqrt{n_k} * Z_{k,score} + \frac{1}{\sqrt{N}} \sum_k \sqrt{n_k} * E_k, \end{aligned} \tag{21}$$

where $E_k \sim N(0, V)$ independently for all k.

Given independence between studies, $\frac{1}{\sqrt{N}} \sum_k \sqrt{n_k} * E_k$ still follows the normal distribution $N(0, V)$. Therefore, the knockoff feature importance statistic at meta-analysis level can be equivalently generated as

$$\widetilde{Z}_{score} = PZ_{score} + E, \text{ where } E \sim N(0, V) \tag{22}$$

The same knockoff filter procedure as before can be applied in this setting.

### Meta-analysis of possibly overlapping studies via knockoffs

The principle for meta-analysis of overlapping studies is to mimic a pooled mega-analysis where the knockoff generation should be revised to account for the presence of possibly duplicated samples. One sufficient condition for a valid knockoff inference is that for a

sample that is present in more than one study its knockoff version for different studies should be identical instead of being independently generated for each study.

Let $N_{effect}$ be the effective number of samples, i.e. the total number of unique samples; $N$ be the total number of records (including duplicates); $d_1, \ldots, d_{N_{effect}}$ be the number of occurrences for each unique sample, $N = d_1 + \ldots + d_{N_{effect}}$. The feature importance score is then defined as

$$Z_{score} = \frac{1}{\sqrt{N}} \sum_k S_k = \frac{1}{\sqrt{N}} \sum_{1 \le i \le N_{effect}} d_i * G_i^T Y_i \qquad (23)$$

Since $\widetilde{G}_i | G_i \sim G_i P^T + e_i^T$ with $e_i \sim N(0, V)$,

$$\widetilde{Z}_{score} = \frac{1}{\sqrt{N}} \sum_k \widetilde{S}_k = \frac{1}{\sqrt{N}} \sum_{1 \le i \le N_{effect}} d_i * \left( P G_i^T Y_i + e_i Y_i \right) \qquad (24)$$

As $e_i$'s are independent for all $i$, in distribution it is equivalent to generate

$$\widetilde{Z}_{score} = \frac{1}{\sqrt{N}} P \sum_k S_k + \sqrt{\frac{\sum_{1 \le i \le N_{effect}} d_i^2 Y_i^2}{N}} E \approx \frac{1}{\sqrt{N}} P \sum_k S_k$$
$$+ \sqrt{\frac{\sum_{1 \le i \le N_{effect}} d_i^2}{N}} E = P Z_{score} + \gamma E \qquad (25)$$

where $E \sim N(0, V)$. The approximation "$\approx$" is because $\frac{\sum_{1 \le i \le N_{effect}} d_i^2 Y_i^2}{N_{effect}}$ is an approximation of $E(d_i^2 Y_i^2) = E(d_i^2) E(Y_i^2) = E(d_i^2)$ since $Y_i$ has mean 0 and variance 1, and $d_i$ and $Y_i$ are independent. Then $E(d_i^2)$ can be approximated by $\frac{\sum_{1 \le i \le N_{effect}} d_i^2}{N_{effect}}$; and we define $\gamma = \sqrt{\frac{\sum_{1 \le i \le N_{effect}} d_i^2}{N}}$, which can be thought of as a "dependency" factor that accounts for sample overlapping. When $d_1 = \ldots = d_{N_{effect}} = 1$, we have $N_{effect} = N$ and $\gamma = 1$, i.e. the scenario with independent studies. It is worth noting that our derivation is based on the following two assumptions: first, different studies are combined by using weights proportional to $\sqrt{n_k / N}$; second, each data point can be observed in different studies with the same probability.

## Calculation of the dependency factor $\gamma$

The number of duplicates per sample is typically unknown. We propose an approximation of $\gamma$ as

$$\gamma = \sqrt{\frac{\sum_{1 \le i \le N_{effect}} d_i^2}{N}} = \sqrt{\frac{N_{effect}}{N} * \frac{\sum_{1 \le i \le N_{effect}} d_i^2}{N_{effect}}} = \sqrt{\frac{N_{effect}}{N} * \overline{d_i^2}} \qquad (26)$$

Under the assumption that the $N - N_{effect}$ duplicates are randomly distributed, $d_i$ follows a distribution $1 + B(N - N_{effect}, \frac{1}{N_{effect}})$, where $B(\cdot)$ denotes a binomial distribution with $N - N_{effect}$ trials and success probability $\frac{1}{N_{effect}}$. Thus

$$\overline{d_i^2} \approx E\left(d_i^2\right) = var(d_i) + E(d_i)^2 = \frac{\left(N - N_{effect}\right)\left(N_{effect} - 1\right)}{N_{effect}^2} + \left(\frac{N}{N_{effect}}\right)^2 \qquad (27)$$

Since $N$ is sufficiently large,

$$\gamma = \sqrt{\frac{N}{N_{effect}} + \left(\frac{N}{N_{effect}} - 1\right)\left(\frac{N_{effect}}{N} - \frac{1}{N}\right)} \approx \sqrt{1 + \frac{N}{N_{effect}} - \frac{N_{effect}}{N}} \qquad (28)$$

## Study correlations and effective sample size

We propose a technique based on the proposed knockoff framework to identify study correlations due to sample overlap. The method requires GWAS summary statistics only and it naturally accounts for LD. Specifically, we first calculate the study correlation matrix

$$cor.S = cor\left(Z_{1,score} - PZ_{1,score}, \ldots, Z_{K,score} - PZ_{K,score}\right) \qquad (29)$$

where $Z_{k,score} - PZ_{k,score} = D\Sigma^{-1} Z_{k,score}$, which is the expected difference between the original Z-score and the knockoff Z-score. It quantifies the putative causal effect adjusting for nearby correlated variants. Under the null hypothesis that genetic variants are independent of the outcome of interest, the correlation between two independent studies is expected to be 0. Thus non-zero off-diagonal elements of $cor.S$ quantifiy the sample overlap. In practice, we use variants with $|Z\text{-score}| \le 1.96$ to calculate $cor.S$ to remove the correlation due to polygenic effects.

We then estimate $N_{effect}$ in a similar way as when estimating effective sample sizes in association studies with sample relatedness[63]. We calculate effective sample size as

$$\hat{N}_{effect} = N * \frac{N}{\sum_{1 \le i,j \le K} \sqrt{n_i n_j} cor.S_{ij}} \qquad (30)$$

where $n_i$ is the sample size of the $i$th study. For example, $\hat{N}_{effect} = N$ if all studies are independent; $\hat{N}_{effect} = N/K$ if all studies are identical. $\frac{N}{\sum_{1 \le i,j \le K} \sqrt{n_i n_j} cor.S_{ij}}$ is the ratio of the variance of $\sum_k S_k$ ignoring sample overlap over that accounting for sample overlap. Thus, we propose an approximation of $\gamma$ as

$$\gamma \approx \sqrt{1 + \frac{N}{\hat{N}_{effect}} - \frac{\hat{N}_{effect}}{N}}, \quad \frac{\hat{N}_{effect}}{N} = \frac{N}{\sum_{1 \le i,j \le K} \sqrt{n_i n_j} cor.S_{ij}} \qquad (31)$$

## Connection with existing meta-analysis methods that allow overlapping samples

A common approach in meta-analysis is to sum Z-scores and weight them properly based on sample sizes, i.e.[16,17].

$$Z_{score} = \frac{1}{\sqrt{N}} \sum_k \sqrt{n_k} * Z_{k,score} \qquad (32)$$

When the Z-scores are independent, $Z_{score}$ follows $N(0,1)$ under the null hypothesis. When there are overlapping samples, the variance is no longer 1. Instead,

$$var(Z_{score}) = \frac{1}{N} \sum_{1 \le i,j \le K} \sqrt{n_i n_j} cov\left(Z_{i,score}, Z_{j,score}\right) \qquad (33)$$

$var(Z_{score}) > 1$ when there are overlapping studies and can be used to compute an effective sample size. Specifically, the $Z_{score}$ should be reduced to

$$Z'_{score} = \sqrt{\frac{N_{eff}}{N}} * \frac{1}{\sqrt{N}} \sum_k \sqrt{n_k} Z_{k,score} \qquad (34)$$

where

$$N_{eff} = \frac{N}{var(Z_{score})} = N * \frac{N}{\sum_{1 \le i,j \le K} \sqrt{n_i n_j} cov\left(Z_{i,score}, Z_{j,score}\right)} \qquad (35)$$

$cov(Z_{i,score}, Z_{j,score})$ quantifies the sample overlap between studies $i$ and $j$, and can be estimated using genome-wide Z-scores. Note that individual Z-scores are normalized, therefore $cov(Z_{i,score}, Z_{j,score}) = cor(Z_{i,score}, Z_{j,score})$. As discussed above, in the proposed approach, we estimate it using the knockoff method that accounts for LD.

## Meta-analysis of overlapping studies with heterogeneous LD structure

Suppose there are $L$ groups (e.g. different ancestries) with different LD structure $\Sigma_1, \ldots, \Sigma_L$, and each group includes $K_l$ overlapping studies with sample size $n_{lk}$ and Z-scores $Z_{lk,score}$; $n_l = \sum n_{lk}$. We assume that each group contains studies with the same LD structure. The overall Z-score is computed as

$$Z_{score} = \frac{1}{\sqrt{N}} \sum_{l,k} \sqrt{n_{lk}} * Z_{lk,score} = \frac{1}{\sqrt{N}} \sum_l \sqrt{n_l} * Z_{l,score} \quad (36)$$

where the Z-score for each group is $Z_{l,score} = \frac{1}{\sqrt{n_l}} \sum \sqrt{n_{lk}} * Z_{lk,score}$. Let $I_{effect,l}$ be the index of unique samples corresponding to the $l$th group, and $N_l$ be its size. The knockoff Z-score can be obtained as

$$\widetilde{Z}_{score} = \frac{1}{\sqrt{N}} \sum_{1 \le i \le N_{effect}} d_i * \widetilde{G}_i^T Y_i = \frac{1}{\sqrt{N}} \sum_{1 \le l \le L} \sum_{i \in I_{effect,l}} d_i * \left( P_l G_i^T Y_i + e_{il} Y_i \right),$$
$$(37)$$

where $e_{il} N(0, V_l)$ independently for all $i$ and $l$. Note that $\sum_{i \in I_{effect,l}} d_i Y_i e_{il}$ still follows a normal distribution $\sqrt{\sum_{i \in I_{effect,l}} d_i^2 Y_i^2} N(0, V_l)$, and in distribution

$$\widetilde{Z}_{score} = \frac{1}{\sqrt{N}} \sum_{1 \le l \le L} \sqrt{N_l} * P_l Z_{l,score} + \frac{1}{\sqrt{N}} \sum_{1 \le l \le L} \sqrt{N_l} * \sqrt{\frac{\sum_{i \in I_{effect,l}} d_i^2 Y_i^2}{N_l}} E_l$$
$$= \frac{1}{\sqrt{N}} \sum_{1 \le l \le L} \sqrt{N_l} * \left( P_l Z_{l,score} + \gamma_l E_l \right) := \frac{1}{\sqrt{N}} \sum_{1 \le l \le L} \sqrt{N_l} * \widetilde{Z}_{l,score}$$
$$(38)$$

where $E_l \sim N(0, V_l)$ independently for all $l$. Thus, the knockoff Z-score can be generated by

$$\widetilde{Z}_{score} = \frac{1}{\sqrt{N}} \sum_{1 \le l \le L} \sqrt{N_l} * \widetilde{Z}_{l,score} \quad (39)$$

which means that it can be generated by the weighted summation of the knockoff Z-scores generated for each individual group.

## Meta-analysis with optimal weights

When meta-analyzing studies with overlapping samples, we would like to down-weight studies that are largely overlapping with others. In general, the proposed meta-analysis approach can be written as

$$Z_{score} = \sum_k w_k Z_{k,score} = \sum_k w_k \sqrt{n_k} * \frac{1}{\sqrt{n_k}} Z_{k,score}. \quad (40)$$

Assuming that the effect sizes per variant from different studies are the same, $\frac{1}{\sqrt{n_k}} Z_{k,score}$ is entry-wise in the same order of $\mu$ where $\mu$ is constant that quantifies the marginal association under the alternative hypothesis. We aim to maximize

$$\frac{Z_{score}}{sd(Z_{score})} = \frac{\sum_k w_k Z_{k,score}}{\sqrt{\sum_{1 \le i,j \le K} w_i w_j cor.S_{ij}}} \sim \frac{\sum_k w_k \sqrt{n_k}}{\sqrt{\sum_{1 \le i,j \le K} w_i w_j cor.S_{ij}}} * \mu \quad (41)$$

which is equivalent to

$$\text{minimize} \sum_{1 \le i,j \le K} w_i w_j cor.S_{ij}, \text{ subject to } \sum_k w_k \sqrt{n_k} = 1, w_k \ge 0. \quad (42)$$

We note that this is similar to the optimal weights proposed by Lin and Sullivan (2009), except for the additional constraint $w_k \ge 0$[17]. It is a convex optimization problem with a unique solution which can be efficiently solved by standard software such as the CVXR package in R. With the proposed weights, we revise the calculation above of the dependency factor as

$$\gamma = \sqrt{1 + \frac{N}{\hat{N}_{eff}} - \frac{\hat{N}_{eff}}{N}}, \quad \frac{\hat{N}_{eff}}{N} = \frac{\sum_k w_k^2}{\sum_{1 \le i,j \le K} w_i w_j cor.S_{ij}} \quad (43)$$

It is worth noting that the form of $\gamma$ was derived for weights $\sqrt{\frac{n_k}{N}}$. Theoretically, the exact form of $\gamma$ should be further revised to reflect the change in weights, which can be complicated to compute in practice. Here we use this form as an approximation, and we found that it empirically controls FDR very well in the presence of sample overlap.

## Practical strategy for study-specific rare variants

Another possible and often overlooked complication in meta-analyses of genetic studies is the unequal coverage of variants across studies due to different genotyping platforms and/or different imputation panels for individual studies, which results in the presence of "study-specific" rare variants (MAF < 0.01) that are measured in only some of the studies. One suboptimal solution is to only include variants measured across all studies. Alternatively, to maximize the power of genetic discovery, all variants may be included in the meta-analyses. In the presence of study-specific rare variants, we propose to calculate the overall Z-score and its knockoff counterpart as

$$Z_{score} = \sum_k w_k * C_k Z_{k,score}, \quad \widetilde{Z}_{score} \sim \sum_k w_k * C_k \left( P Z_{k,score} + \gamma E_k \right) \quad (44)$$

where $C_k$ is a diagonal matrix with $c_{kj} = 1$ if the $j$th variant is observed in study $k$ and $c_{kj} = 0$ otherwise; $E_k \sim N(0, V)$ independently for all $k$. Intuitively, we generate modified knockoff Z-scores $P Z_{k,score} + \gamma E_k$ for each study and combine them as one meta-analysis knockoff Z-score. When a variant is not measured in a study, we propose coding both its Z-score and knockoff Z-score for that study as 0. This way the study does not contribute to the meta-analysis Z-score/knockoff Z-score. It is worth noting that knockoff inference is scale-free because the feature selection is based on a contrast between $Z_{score}$ and $\widetilde{Z}_{score}$. Therefore, it does not require rescaling $Z_{score}$ and $\widetilde{Z}_{score}$ to account for the reduced variation due to study-specific variants.

## Practical strategy for tightly linked variants

Although the knockoff method helps to prioritize causal variants over associations due to LD, it is difficult or impossible to distinguish causal genetic variants from highly correlated variants. The presence of tightly linked variants can diminish the power to identify the causal ones. We applied a hierarchical clustering of genetic variants prior to the analysis, where variants in the same cluster have a pair-wise correlation ≥0.75. Then we restricted the analysis to one randomly selected representative variant in each cluster. This ensures that each genetic variant in the genome has a highly correlated representative to be included in the analysis and the analysis is unbiased. On average (based on the nine AD studies), we observed that 37.9% variants can be matched with the 1000 Genome reference panel for the proposed GhostKnockoff analysis after this pruning procedure. That is, on average, each variant represents 2.64 variants in the same cluster.

## Empirical power and FDR simulations

For each replicate, we first generated two overlapping studies (2500 individuals per study) with genetic data on 2000 common and rare genetic variants randomly selected from a 1Mb region near the *APOE* region (chr19:44909011-45912650; hg38) in the ADSP study. We then restricted the simulations to variants with minor allele counts >25 to ensure stable calculation of summary statistics (e.g. *p*-values). Since the simulations here focus on method comparison to identify relevant clusters of tightly linked variants, we simplify the simulation design by keeping one representative variant from each tightly linked cluster. Specifically, we applied hierarchical clustering such that no two clusters have cross-correlations above a threshold value of 0.75 and then randomly choose one representative variant from each cluster to be included in the simulation study. To simulate multiple causal variants, we randomly set 10 variants in the 1Mb region to be causal, with a positive effect on the quantitative/dichotomous trait as follows:

$$\text{Quantitative trait}: Y_i = X_{i1} + \beta_1 G_{i,1} + \ldots + \beta_{10} G_{i,10} + \varepsilon_i^Q \quad (45)$$

$$\text{Dichotomous trait}: g(\mu_i) = \beta_0 + X_{i1} + X_{i2} + \beta_1 G_{i,1} + \ldots + \beta_{10} G_{i,10} \quad (46)$$

where $X_{i1} \sim N(0,1)$, $\varepsilon_i^Q \sim N(0,3)$, $X_{i2} \sim N(0,1)$ and they are all independent; $X_{i1}$ is the observed covariate that is adjusted in the analysis; $\varepsilon_i^Q$ and $X_{i2}$ reflect variation due to unobserved covariates; $(G_{i,1}, \ldots, G_{i,10})$ are selected risk variants; $g(x) = \log(\frac{x}{1-x})$ and $\mu_i$ is the conditional mean of $Y_i$; for dichotomous trait, $\beta_0$ is chosen such that the prevalence is 10%. We set the effect $\beta_j = \frac{a}{\sqrt{2m_j(1-m_j)}}$, where $m_j$ is the MAF for the $j$th variant. We define $a$ such that the variance due to the risk variants, $\beta_1^2 var(g_1) + \ldots + \beta_{10}^2 var(g_{10})$, is 1. We applied the proposed methods to the region as described before to analyze single variants. For each replicate, the empirical power is defined as the proportion of detected variants among all causal variants; the empirical FDR is defined as the proportion of non-causal variants among all detected variants. We simulated 1000 replicates and calculated the average empirical power and FDR.

## Reporting summary

Further information on research design is available in the Nature Portfolio Reporting Summary linked to this article.

## Data availability

The manuscript used summary statistics from existing studies from the UK Biobank available at https://pheweb.org/UKB-SAIGE/. The summary statistics from each GWAS for Alzheimer's disease can be found at https://ctg.cncr.nl/software/summary_statistics, https://www.niagads.org/datasets, and https://www.ebi.ac.uk/gwas/. Specifically, (1) The genome-wide survival association study performed by Huang et al. 2017 (NIAGADS ID: NG00058); (2) The genome-wide meta-analysis by Jansen et al. 201944 (available through: https://ctg.cncr.nl/software/summary_statistics); (3) The genome-wide meta-analysis by Kunkle et al. 2019 (NIAGADS ID: NG00075); (4) The genome-wide meta-analysis by Schwartzentruber et al. 2021 (GWAS catalog ID: GCST90012877); (5) In-house genome-wide associations study imputed using the TOPMed reference panels (see Supplementary Table 3); (6,7) Two whole-exome sequencing analyses of data from ADSP by Bis et al. 2020 (NIAGADS ID: NG00065), and Le Guen et al. 2021 (NIAGADS ID: NG000112); (8) In-house whole-exome sequencing analysis of ADSP (NIAGADS ID: NG00067.v5); (9) In-house whole-genome sequencing analysis of ADSP (NIAGADS ID: NG00067.v5). The single-cell RNASeq data for the candidate genes are available in the GEO database under accession code GSE163577. The results of our analysis of UK Biobank and AD genetics can be downloaded at: zihuaihelab.github.io.

## Code availability

We have implemented GhostKnockoff in a computationally efficient R package that can be accessed at https://cran.r-project.org/web/packages/GhostKnockoff/.

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

## Acknowledgements

This research was additionally supported by NIH/NIA award AG066206 (Z.H.), AG060747 (M.D.G.), AG066515 (Z.H., T.W.-C., M.D.G.). We gratefully acknowledge the studies which provided summary statistics.

## Author contributions

Z.H. developed the concepts for the manuscript and proposed the method. Z.H. L.L., H.T. M.G., and I.I.-L designed the analyses and

applications and discussed results. L.L., C.S., E.C. significantly improved the mathematical rigor of the method. Z.H., M.E.B., Y.L.G., A.S., X.L., X.Q., P.G., and S.M conducted the analyses. M.E.B. and M.D.G. helped with design and interpretation of AD genetics. T.W.C. helped with design and interpretation of the scRNAseq analyses. Z.H. prepared the manuscript and all authors contributed to editing the paper.

## Competing interests

The authors declare no competing interests.
