## [Peer Review File · Nature Communications]

GhostKnockoff inference empowers identification of putative causal variants in genome-wide association studiesREVIEWER COMMENTS

Reviewer #1 (Remarks to the Author):

He et al propose GhostKnockoff, a method for performing feature selection in GWAS using only summary statistics (and a reference panel). This work extends recent work that uses the "knockoff" technique but with individual-level data, resulting in a method that is more applicable (because in many cases, it is not possible to obtain individual-level data) and faster.

I am impressed by the methods of this paper. I had not previously heard of the knockoff technique (to which end, I first read existing literature on the method, including this helpful article <https://www.stat.cmu.edu/~ryantibs/journalclub/knockoff.pdf>, and partly read the authors previous work <https://www.nature.com/articles/s41467-021-22889-4> of which this is an extension). While extending from individual-level data to summary statistics is usually fairly straightforward, I believe this is the first example in this setting (i.e., for knockoffs, which are quite different to "conventional analyses"). Further, the authors appear thorough in their consideration of special cases (e.g., a method of meta-analysis for cohorts with overlapping samples, that appears novel).

However, I am less impressed by the application. The main challenge is showing their results are meaningful - i.e., that the associations they declare are correct (true) or useful. However, this paper's first comparisons (from UKBB) are hard to interpret because they use False Discovery Rate (FDR) to define significant associations from their method, but FWER to define associations from the standard approach. The former is much less strict - I estimate the FWER threshold corresponds to a FDR of about $0.05/50 = 0.001$, much lower than the threshold of 0.1 they use for their new method (while the authors recognise this difference, it remains an important issue in my view). The authors argue that their results are more interesting because they are "more independent" (46% less variants within each locus), but I find this hard to appreciate. Saying this, I do appreciate that the nature of results from knockoffs and conventional GWAS are systematically different, which means it is possible the former are providing new information.

Then for the AD example, it seems that the novel associations could mainly be because they are using more samples (they combine results from nine cohorts, then compare to results from only three cohorts, albeit I believe the largest three), instead of due to better methods. I appreciate they described the function / Phred score of the novel associations, although I see this as quite speculative.

My personal strategy to demonstrate the method would be to apply your method to a subset of samples, and show that the hits you find are validated (also found or are marginally significant) when you increase the sample size (or use a larger, independent sample). This typically requires individual-

level data (e.g., UKBB data, so that you can create summary statistics corresponding to different subsets of individuals). Otherwise, you could instead use pseudo-summary statistics (see below).

So in summary, I like the methods, but did not think the application was strong, because the comparisons were of fundamentally different methods (ie FDR vs FWER correction), and there was no attempt to replicate novel associations.

Major comments

1 - As I say above, the methods are generally convincing, but the application less so.

2 - Many summary statistic methods can suffer, for example, when there are regions of high LD (eg the MHC). Was this an issue? More generally, were there ever convergence problems (eg with the optimization of the weightings?) or other issues common to summary statistics (eg mismatch of reference panel).

Minor comments

3 - In the methods, I believe Sigma is the SNP-SNP correlation matrix across the whole genome. But this is impractical to obtain (compute a 1M x 1M matrix). How therefore do you go about this? Do you divide the genome into blocks? Or is there a way to avoid explicitly computing Sigma. Further, for the multiple knockoffs, it seemed you had a pM x pM matrix (so M times bigger than above). How is this feasible?

4 - The definition of independent loci (>200kb) is very unusual. A more common definition is 1Mb, or 1cM, or 3Mb/3cm (usually with an r^2 threshold, of say 0.1).

5 - I do not approve of reporting times across multiple CPUs - in my view, it is harder for fair comparison and can give misleading impression of time. In my view, it would be better to report times for one CPU (then you could add speed up through parallelization in brackets)

6 - Page 3, Line 10 - p (no of snps) is not defined. Further, please define this (and other symbols) at the start of the methods (so that the methods can be standalone)

7 - Could you perhaps summarize the advantage of knockoffs over making null test statistics, by sampling Z (tilda) score from $N(0, R)$, where R is SNP-SNP correlation matrix (or equiv, the advantage of knockoffs with individual-level data versus permutation analyses).

8 - P8 - L 14 "We applied GhostKnockoff separately to each of the 1,403 binary phenotype" - I think this should be "each of 1,403" (you have 1403 binary phenotypes, but it is not the case that UKBB contains exactly 1,403 phenotypes - sorry if this is a bit pedantic). Further, your paper text gave the impression you ran SAIGE yourself, which would be impressive, but only in data availability is it clear you instead used results from another group.

9 - P8 - L 20 "Due to the relatively small sample size of the 1000 Genomes Project (503 individuals of European ancestry) used to estimate the LD matrices, we restrict the analyses to common and low-frequency variants with minor allele frequency $>1\%$." While I am fine with your threshold, is there ever a situation you would use lower threshold (as in, regardless of the sample size of the ref panel, can you ever confidently estimate LD for rare variants?)

10 - On figure 4b, it would be nice to see the standard weights (prop to $1/\text{sample size}$ for each study) as a comparison (ie to see how much difference your clever weighting scheme makes).

11 - In general, I did not like that the results section contained primarily methods. I realise that given the journal's format requirement (that methods should be at the end), it becomes necessary to insert some methods in the results, but usually, this would be a quick summary (not five or six pages). However, the editor can advise.

12 - It seems this knockout approach has similarities with my "pseudo summary statistics" (<https://www.nature.com/articles/s41467-021-24485-y>), which create fake summary statistics (equiv to z scores), but with less association with outcome, instead of none. Please note, I may well be wrong, and there is no similarity, or the similarity is slight / inconsequential, in which case, you are certainly not required to discuss the similarity in the paper.

13 - The extension to overlapping samples is nice

14 - Can you provide an explanation (perhaps only to me) of the use of the weights s_j . It seems their introduction is to ensure V is positive definite. Further are there ever problems finding a solution to the optimization problem. For example, does the method suffer if only able to find a poor solution.

Signed Doug Speed

Sorry that I am slightly slower to submit this review than I hoped (23 days).

Reviewer #2 (Remarks to the Author):

The authors propose a novel method, GhostKnockoff, that leverages a knockoff inference framework to prioritize putative causal variants at a locus using summary statistics. This method accounts for linkage disequilibrium patterns and can generate knockoff feature importance scores without the generation of individual-level knockoffs for improved computational efficiency. The authors demonstrate the performance of their method using data on over one thousand phenotypes from the UK biobank and a separate meta-analysis of Alzheimer's disease studies. They identified 55 AD loci (31 of which appear to be novel) and significantly improved statistical power compared to conventional GWAS. Overall, this appears to be a powerful new addition to the genetics toolbox and I have only minor queries/suggestions.

Minor suggestions

1. Pooling whole genome data with whole exome data as well as combining multiple overlapping studies requires a lot of careful technical consideration. The ability to combine these types of datasets using a single tool at the summary stats level sounds nearly too good to be true! How do differences in data generation, genotype coverage, and imputation impact the validity of inferred study overlap and subsequent analytical results?
2. It would be great to see a PRS performance prediction comparison between GhostKnockoff and one of the individual summary stats in the AD meta-analysis section. It would be compelling to see improved performance using your novel method.

3. Take this as a not required but greatly-appreciated-if-addressed suggestion: Something that would add even more value to this manuscript is a proof-of-concept assessment of two GWAS with different LD patterns (e.g., from different genetic ancestries) to highlight the flexibility and increased power of GhostKnockoff in this type of scenario.

Other small issues:

-Line 14 page 3: explain what type of data was used from UK Biobank (WGS) to be consistent with the data description of the meta-analysis

-APOE e4 status in the single-cell section should be reported and assessed with relation to the DEG findings (previous studies have shown that there is an association between differentially expressed genes and APOE e4 status)

-It would be good for the authors to mention the caveat that the most proximal gene is not necessarily the most likely [functional] candidate

-Chr 22 label is missing in Figure 4

We thank the reviewers for their detailed and insightful comments. In the revised version, we have made several important modifications to address these concerns. Specifically, we have clarified the main difference between knockoff inference and the conventional GWAS, revised the analysis sections, and changed the interpretation of the results. We have also added additional analyses to validate the results from the knockoff inference, using a strategy suggested by reviewer #1. Finally, we have expanded upon the limitations in the Discussion section to address the comments.

Reviewer #1 (Remarks to the Author):

He et al propose GhostKnockoff, a method for performing feature selection in GWAS using only summary statistics (and a reference panel). This work extends recent work that uses the "knockoff" technique but with individual-level data, resulting in a method that is more applicable (because in many cases, it is not possible to obtain individual-level data) and faster.

I am impressed by the methods of this paper. I had not previously heard of the knockoff technique (to which end, I first read existing literature on the method, including this helpful article <https://www.stat.cmu.edu/~ryantibs/journalclub/knockoff.pdf>, and partly read the authors previous work <https://www.nature.com/articles/s41467-021-22889-4> of which this is an extension). While extending from individual-level data to summary statistics is usually fairly straightforward, I believe this is the first example in this setting (i.e., for knockoffs, which are quite different to "conventional analyses"). Further, the authors appear thorough in their consideration of special cases (e.g., a method of meta-analysis for cohorts with overlapping samples, that appears novel).

However, I am less impressed by the application. The main challenge is showing their results are meaningful - i.e., that the associations they declare are correct (true) or useful. However, this paper's first comparisons (from UKBB) are hard to interpret because they use False Discovery Rate (FDR) to define significant associations from their method, but FWER to define associations from the standard approach. The former is much less strict - I estimate the FWER threshold corresponds to a FDR of about $0.05/50 = 0.001$, much lower than the threshold of 0.1 they use for their new method (while the authors recognise this difference, it remains an important issue in my view). The authors argue that their results are more interesting because they are "more independent" (46% less variants within each locus), but I find this hard to appreciate. Saying this, I do appreciate that the nature of results from knockoffs and conventional GWAS are systematically different, which means it is possible the former are providing new information.

Then for the AD example, it seems that the novel associations could mainly be because they are using more samples (they combine results from nine cohorts, then compare to results from only three cohorts, albeit I believe the largest three), instead of due to better methods. I appreciate they described the function / Phred score of the novel associations, although I see this as quite speculative.

My personal strategy to demonstrate the method would be to apply your method to a subset of samples, and show that the hits you find are validated (also found or are marginally significant) when you increase the sample size (or use a larger, independent sample). This typically requires individual-level data (e.g., UKBB data, so that you can create summary statistics corresponding to different subsets of individuals). Otherwise, you could instead use pseudo-summary statistics (see below).

So in summary, I like the methods, but did not think the application was strong, because the comparisons were of fundamentally different methods (ie FDR vs FWER correction), and there was no attempt to replicate novel associations.

Response: Thank you very much for the nice summary, and the helpful comments. Below please find our responses to your comments.

Major comments:

1 - As I say above, the methods are generally convincing, but the application less so.

Response: We agree with the reviewer that the proposed knockoff inference is systematically different from the conventional GWAS. Specifically, conventional GWAS performs a marginal association test and controls for FWER. It tests against hypothesis

$$H_0: Y \perp G_j \text{ for } j = 1, \dots, p$$

where Y is the outcome; G_j is the j -th genetic variant. The knockoff inference performs a conditional test and controls for FDR. It tests against hypothesis

$$H_0: Y \perp G_j | \mathbf{G}_{-j} \text{ for } j = 1, \dots, p$$

where \mathbf{G}_{-j} includes all genetic variants except the j -th variant.

Marginal test vs. conditional test. Marginal test often identifies a cluster of proxy variants at the same locus that are correlated with the true causal variant. Conditional on the true causal variant, the proxy variants are no longer associated with the outcome. The knockoff inference performs conditional tests that account for linkage disequilibrium (LD) thereby reducing false positive findings due to LD confounding at each locus. Consequently, we showed in the UK biobank analysis that the number of proxy variant at each locus is reduced by using knockoff inference.

FWER vs. FDR. Although we are not sure we follow the estimation derived by reviewer (0.05/50) for the connection between the two, we agree that the FDR control is indeed more liberal compared to FWER. The knockoff inference adopts the FDR control to identify small effect risk loci that currently lie below the genome-wide significance threshold even in large GWAS. We demonstrated this property in the UK biobank analysis, where the number of identified loci is increased. We also showed that the additional loci are informative by functional enrichment analysis. However, we do acknowledge that the increase in the number of loci is due to a more liberal criterion (note that the knockoff provide control of FDR in presence of arbitrary correlations, while the conventional FDR procedures do not).

In the new manuscript, we have clarified the difference in methods (Page 3, line 32), revised the analysis sections, and changed the interpretation of the results. Particularly, we have rephrased the comparison as a way to explain the difference, instead of demonstrating that one is superior to the other (Page 10, line 26).

In addition, we have added additional analysis to validate the results from the knockoff inference (Page 9, line 14). We adopted the strategy suggested by the reviewer, i.e. to apply the method to a subset of samples, and show that the identified variants are also found when we increase the sample size. Specifically, we considered GhostKnockoff analysis of Kunkle et al. (2019), Schwartzentruber et al. (2021), and all nine studies. Note that data from Kunkle et al. (2019) is a subset of Schwartzentruber et al. (2021); Schwartzentruber et al. (2021) is a subset of the meta-analysis. We considered replication of genetic variants. A genetic variant is replicated if the same variant is also identified in the larger study with the same direction of effect. We present the results for FDR=0.05 and FDR=0.10 in Table 1 (Table 1 in this letter) and Supplementary Figure 3 (Figure 1 in this letter).

At FDR=0.05, we observed that 345 out of 385 (89.6%) variants by GhostKnockoff analysis of Kunkle et al. (2019) are also identified in the analysis of Schwartzentruber et al. (2021); 525 out of 634 (83%) variants identified by GhostKnockoff analysis of Schwartzentruber et al. (2021) are also identified in the proposed meta-analysis of all nine studies. At FDR=0.10, 378 out of 448 (84.4%) variants identified by GhostKnockoff analysis of Kunkle et al. (2019) are also identified in the analysis of Schwartzentruber et al. (2021); 599 out of 724 (82.7%) variants identified by GhostKnockoff analysis of Schwartzentruber et al. (2021) are also identified in the proposed meta-analysis of all nine studies. Overall, we conclude that the proposed GhostKnockoff (conditional test + FDR control), though systematically different from the conventional GWAS, is a valid approach to make reproducible genetic discoveries.

Table 1: Replication of variants identified by GhostKnockoff in larger studies. The analysis reflects the application of the proposed method to a subset of samples and the validation of the findings when we increase the sample size. We present the number of identified variants by applying GhostKnockoff to summary statistics from Kunkle et al. (2019), Schwartzentruber et al. (2021) (a study aggregating samples from Kunkle et al. (2019) and UK Biobank based on a proxy AD phenotype), and the proposed GhostKnockoff meta-analysis based on the optimal weights combining nine studies. A genetic variant is replicated if the same variant is also identified in the next larger study with the same direction of effect.

		Source of summary statistics		
		Kunkle et al. (2019)	Schwartzentruber et al. (2021)	Proposed meta-analysis of all nine studies
FDR=0.05	Total number	385	634	764
	Number (proportion) of replicated discoveries	345 (89.6%)	525 (83%)	-
FDR=0.10	Total number	448	724	935
	Number (proportion) of replicated discoveries	378 (84.4%)	599 (82.7%)	-

Figure 1: Replication of variants and loci identified by GhostKnockoff in larger studies. The analysis reflects the application of the proposed method to a subset of samples and the validation of the findings when we increase the sample size. We present the Manhattan plot of W statistics (truncated at 100 for clear visualization) from GhostKnockoff with target FDR at 0.05 (red dotted line; loci are highlighted in red) and 0.10 (blue dotted line; loci are highlighted in blue). **A.** GhostKnockoff analysis based on summary statistics from Kunkle et al. (2019). **B.** GhostKnockoff analysis based on summary statistics from Schwartzentruber et al. (2021), a study aggregating samples from Kunkle et al. (2019) and UK Biobank based on a proxy AD phenotype. **C.** The proposed GhostKnockoff meta-analysis based on the optimal weights combining nine studies. Variant density is shown at the bottom of Manhattan plot (number of variants per 1Mb).

2 - *Many summary statistic methods can suffer, for example, when there are regions of high LD (eg the MHC). Was this an issue? More generally, were there ever convergence problems (eg with the optimization of the weightings?) or other issues common to summary statistics (eg mismatch of reference panel).*

Response: We appreciate the reviewer's thoughtful comments. For the issue related to high LD region, we think that it is certainly a limitation of the proposed method as already discussed in the subsection "Practical strategy for tightly linked variants" and in the Discussion section. We expect that the proposed method won't be able to distinguish the causal variant from the proxy ones in high LD regions like the MHC. We have further added this to the Discussion section (page 12, line 4).

For the potential convergence problems raised by the reviewer, particularly with the optimization of the weighting in the meta-analysis, we clarify that it is a convex optimization problem with a unique solution which can be efficiently solved by standard software such as the CVXR package in R. We did not experience any convergence issue. We have added this information to the paper (page 24, line 24).

For the mismatch of reference panel, we agree that this is a potential issue, similar to other summary statistics-based methods. The current method assumes a matched reference panel to ensure the equivalence between knockoff inference based on individual level data and GhostKnockoff. We have discussed this in the Discussion section (page 11, line 33).

Minor comments:

3 - *In the methods, I believe Sigma is the SNP-SNP correlation matrix across the whole genome. But this is impractical to obtain (compute a $1M \times 1M$ matrix). How therefore do you go about this? Do you divide the genome into blocks? Or is there a way to avoid explicitly computing Sigma. Further, for the multiple knockoffs, it seemed you had a $pM \times pM$ matrix (so M times bigger than above). How is this feasible?*

Response: Thank you for the valuable comment. It is not feasible to work with the genome-wide correlation matrix. In practice, we divided the genome into blocks that can be loaded into memory and performed the intermediate calculation of the knockoff statistics. Then the knockoff statistics from all blocks are aggregated for a genome-wide feature selection. This practical solution does not model inter-block correlation. Consequently, the current method cannot attenuate the confounding effect of long-range LD. We have added this limitation to the Method section (Page 19, line 20)

4 - *The definition of independent loci ($>200kb$) is very unusual. A more common definition is $1Mb$, or $1cM$, or $3Mb/3cM$ (usually with an r^2_{jk} threshold, of say 0.1).*

Response: We have changed the definition of independent loci to $1Mb$ and updated the analysis results.

5 - *I do not approve of reporting times across multiple CPUs - in my view, it is harder for fair comparison and can give misleading impression of time. In my view, it would be better to report times for one CPU (then you could add speed up through parallelization in brackets).*

Response: We have made the change to report time based on one CPU (Page 3, line 20; Page 6, line 12). For the AD meta-analysis, we compute the time for chromosome 1 based on one CPU and then extrapolated the time for the genome based on the chromosome size. For the UK biobank data, it was originally based on one CPU per phenotype. So we simply made the change to report that time.

6 - *Page 3, Line 10 - p (no of snps) is not defined. Further, please define this (and other symbols) at the start of the methods (so that the methods can be standalone).*

Response: We have made the changes.

7 - *Could you perhaps summarize the advantage of knockoffs over making null test statistics, by sampling Z (tilda) score from $N(0,R)$, where R is SNP-SNP correlation matrix (or equiv, the advantage of knockoffs with individual-level data versus permutation analyses).*

Response: The main difference is that permutation analysis still tests against hypothesis

$$H_0: Y \perp G_j \text{ for } j = 1, \dots, p$$

while the knockoff inference tests against hypothesis

$$H_0: Y \perp G_j | \mathbf{G}_{-j}$$

This way knockoff inference attenuates the confounding effect of LD. We have clarified this in the manuscript (Page 4, line 1).

8 - *P8 - L 14 "We applied GhostKnockoff separately to each of the 1,403 binary phenotype" - I think this should be "each of 1,403" (you have 1403 binary phenotypes, but it is not the case that UKBB contains exactly 1,403 phenotypes - sorry if this is a bit pedantic). Further, your paper text gave the impression you ran SAIGE yourself, which would be impressive, but only in data availability is it clear you instead used results from another group.*

Response: We have made the changes. In addition, we have cited the source of SAIGE summary statistics.

9 - *P8 - L 20 "Due to the relatively small sample size of the 1000 Genomes Project (503 individuals of European ancestry) used to estimate the LD matrices, we restrict the analyses to common and low-frequency variants with minor allele frequency >1%." While I am fine with your threshold, is*

there ever a situation you would use lower threshold (as in, regardless of the sample size of the ref panel, can you ever confidently estimate LD for rare variants?)

Response: We agree that it is more challenging to confidently estimate LD for rare variants. However, we think that the valid estimation of LD depends on the frequency and the sample size of the reference panel. For example, recent studies have shown that imputation quality of rare variants can be significantly improved by using better reference panels with larger sample sizes, e.g. the TOPMed reference (Figure 2 in this letter). We believe that we will be able to study variants with minor allele frequency <1% in the future. Certainly, the estimation of LD will be less accurate for variants with lower frequency. We have expanded our discussion to address this issue (Page 11, line 41).

Figure 2. Imputation quality of different panels.

Source Michigan imputation server:

<https://twitter.com/umimpute/status/1248365773850136578/photo/1>

10 - On figure 4b, it would be nice to see the standard weights (prop to 1/sample size for each study) as a comparison (ie to see how much difference your clever weighting scheme makes).

Response: We have added the results based on standard weights (prop to 1/sample size) (Supplementary Figure 4; Figure 3 in this letter). We observed that GhostKnockoff with Standard weights identified 29 loci at FDR 0.05 and 46 loci at FDR 0.10, while GhostKnockoff with proposed weighting scheme identified 34 loci at FDR 0.05 and 50 loci at FDR 0.10.

Figure 3: Comparison between sample size weights (top panel; 29 loci at FDR 0.05; 46 loci at FDR 0.10) vs. optimal weights (bottom panel; 34 loci at FDR 0.05; 50 loci at FDR 0.10). We present the Manhattan plot of W statistics (truncated at 100 for clear visualization) from GhostKnockoff with target FDR at 0.05 (red) and 0.10 (blue). Variant density is shown at the bottom of Manhattan plot (number of variants per 1Mb).

11 - In general, I did not like that the results section contained primarily methods. I realise that given the journal's format requirement (that methods should be at the end), it becomes necessary to insert some methods in the results, but usually, this would be a quick summary (not five or six pages). However, the editor can advise.

Response: Thank you for the suggestion. We would be happy to shorten this section if the editor finds it appropriate.

12 - It seems this knockout approach has similarities with my "pseudo summary statistics" (<https://www.nature.com/articles/s41467-021-24485-y>), which create fake summary statistics (equiv to z scores), but with less association with outcome, instead of none. Please note, I may well be wrong, and there is no similarity, or the similarity is slight / inconsequential, in which case, you are certainly not required to discuss the similarity in the paper.

Response: We carefully read the suggested paper. We think that the pseudo summary statistics approach is very interesting and allows one to validate a prediction model without individual level data. However, we feel that it is different from the knockoff generation which requires strong exchangeability conditions.

13 - The extension to overlapping samples is nice.

Response: Thanks a lot.

14 - Can you provide an explanation (perhaps only to me) of the use of the weights s_j . It seems their introduction is to ensure V is positive definite. Further are there ever problems finding a solution to the optimization problem. For example, does the method suffer if only able to find a poor solution.

Response: Intuitively, the s_j values quantify the distance between original feature and knockoffs. Since the feature selection is based on the contrast between the original feature and its knockoffs, the power of feature selection is related to the magnitude of s_j . Therefore, the algorithm aims to maximize s_j while ensuring the exchangeability condition. It solves a semidefinite programming problem which usually has a unique solution. If it numerically ends up with a poor solution (i.e. $s_j = 0$), there will be no power to identify the corresponding variant.

Reviewer #2 (Remarks to the Author):

The authors propose a novel method, GhostKnockoff, that leverages a knockoff inference framework to prioritize putative causal variants at a locus using summary statistics. This method accounts for linkage disequilibrium patterns and can generate knockoff feature importance scores without the generation of individual-level knockoffs for improved computational efficiency. The authors demonstrate the performance of their method using data on over one thousand phenotypes from the UK biobank and a separate meta-analysis of Alzheimer's disease studies. They identified 55 AD loci (31 of which appear to be novel) and significantly improved statistical power compared to conventional GWAS. Overall, this appears to be a powerful new addition to the genetics toolbox and I have only minor queries/suggestions.

Minor suggestions:

1. Pooling whole genome data with whole exome data as well as combining multiple overlapping studies requires a lot of careful technical consideration. The ability to combine these types of datasets using a single tool at the summary stats level sounds nearly too good to be true! How do differences in data generation, genotype coverage, and imputation impact the validity of inferred study overlap and subsequent analytical results?

Response: Thank you for this valuable comment. We focus on sample overlap because it is one of the main sources of study correlation, but the method can be more general to quantify study correlation due to other factors such as data generation, genotype coverage, imputation, and phenotype definition. In fact, the proposed estimation of study correlation is valid as long as the correlation of genome-wide Z-scores is correctly inferred (this is reflected in the derivations in section “Study correlations and effective sample size”). Our proposed method for estimating the correlation is data driven. If other factors increase/decrease the correlation, we think that the data-driven estimation of the study correlation will remain valid. However, we do require that p-values from different datasets are valid, and correctly reflecting the same disease-genetics association. If the p-values in the original study are deflated or inflated, the meta-analysis results can be biased. In addition, the dependency factor for knockoff generation was derived based on sample overlapping. In practice, we found that the analysis based on this dependency factor reasonably reflects other factors, but the theoretical guarantee will require future investigations. We have added this to the Discussion (Page 12, line 15).

2. It would be great to see a PRS performance prediction comparison between GhostKnockoff and one of the individual summary stats in the AD meta-analysis section. It would be compelling to see improved performance using your novel method.

Response: We agree that it is a very interesting question. We think that the improvement can be by leveraging more independent loci and by accounting for the LD confounding effect. However, we currently do not have a hold-out dataset that is independent of all studies considered in this paper to test the PRS constructed via GhostKnockoff. It will be important to study this in the future.

3. Take this as a not required but greatly-appreciated-if-addressed suggestion: Something that would add even more value to this manuscript is a proof-of-concept assessment of two GWAS with different LD patterns (e.g., from different genetic ancestries) to highlight the flexibility and increased power of GhostKnockoff in this type of scenario.

Response: We thank the reviewer for this suggestion. We are indeed working on a separate paper to focus on GhostKnockoff inference of multiple ancestry groups, and particularly to improve its performance for minority populations.

Other small issues:

-Line 14 page 3: explain what type of data was used from UK Biobank (WGS) to be consistent with the data description of the meta-analysis.

Response: We have made the changes.

-APOE e4 status in the single-cell section should be reported and assessed with relation to the DEG findings (previous studies have shown that there is an association between differentially expressed genes and APOE e4 status).

Response: We agree that APOE e4 status is important in the analysis of Alzheimer's disease transcriptomics. However, the current single-cell dataset does not include the APOE status as part of the study.

-It would be good for the authors to mention the caveat that the most proximal gene is not necessarily the most likely [functional] candidate.

Response: We have clarified this in the relevant section (Page 8, line 42)

-Chr 22 label is missing in Figure 4.

Response: We have added the label.

REVIEWER COMMENTS

Reviewer #1 (Remarks to the Author):

In general, the authors have replied carefully to my comments. However, I have two moderately negative comments.

I appreciate the authors adding Table 1 (replication at different sample sizes). However, this does not seem to demonstrate replication. So for example, you find that 448 SNPs are significant for sample size "1" (Kunkle), of which 378 are significant for sample size "2" (Schwartzentruber). I can't tell from these numbers if this is good. To give an example, suppose a SNP has positive effect and p-value $1e-6$ for sample size "1", then positive effect and p-value $1e-5$ for sample size "2". Your table would count this as a "replication", but actually this scenario is consistent with the SNP being a false positive for sample size "1" (due to the fact there is sample overlap).

I'm sorry if my original comment was not clear. The standard way to replicate would be to analyze for Kunkle, then analyze for the samples in Schwartzentruber but not in Kunkle (i.e., a dataset independent of Kunkle), then see if the significant SNPs from the first are also significant in the second analysis. I appreciate this is hard / impossible, because you can not obtain the second dataset (and even if you could, it might be so small you don't have power to replicate).

Therefore, my suggestion would be to do what you did, but for the SNPs significant for sample size "1", report the proportion that are more significant for sample size "2" (and in the same direction, as you already have done).

Ideally, the proportion would be high (and if you wished, you could test if it was significantly greater than 0.5).

While I was happy with the responses to my other comments (particularly my very technical ones), I was not happy with the response to R2, Point 3. The authors claim to have carefully read my paper (their

response to my Point 12), and yet that paper exactly provides a way to do the PRS analysis R2 suggested (by allowing them to generate a pseudo independent dataset. As in, they can divide the summary statistics from all samples into pseudo training ss and pseudo test ss, then use the former to construct the PRS, and the latter to test...

Lastly, it seems from the response, that the editor has not yet responded to my Point 11.

Signed, Doug Speed

ps, I sign my name both in an effort to improve transparency of the review process, and because I am happy for authors to contact me with queries (it seems more efficient than only allowing communication between authors and reviewers through submissions). Therefore, if you would like clarification, please email me.

Reviewer #2 (Remarks to the Author):

Thank you for addressing my questions. I have no additional concerns.

We thank the reviewer #1 once again for their thorough examination of our manuscript and for his additional comments. In the revised version, we have made modifications to address these concerns by updating the validation of the results from the knockoff inference. We also appreciate the comment and suggestion on the PRS analysis which was originally raised by reviewer #2. We curated additional individual-level datasets and performed analyses to address the comment, although we do consider it to be slightly outside the scope of the current manuscript which focuses on feature selection instead of prediction.

Reviewer #1 (Remarks to the Author):

In general, the authors have replied carefully to my comments. However, I have two moderately negative comments.

1. I appreciate the authors adding Table 1 (replication at different sample sizes). However, this does not to me seem to demonstrate replication. So for example, you find that 448 SNPs are significant for sample size "1" (Kunkle), of which 378 are significant for sample size "2" (Schwartzentruber). I cant tell from these numbers if this is good. To give an example, suppose a SNP has positive effect and p-value $1e-6$ for sample size "1", then positive effect and p-value $1e-5$ for sample size "2". Your table would count this as a "replication", but actually this scenario is consistent with the SNP being a false positive for sample size "1" (due to the fact there is sample overlap).

Im sorry if my original comment was not clear. The standard way to replicate would be to analyze for Kunkle, then analyze for the samples in Schwartzentruber but not in Kunkle (i.e., a dataset independent of Kunkle), then see if the significant SNPs from the first are also significant in the second analysis. I appreciate this is hard / impossible, because you can not obtain the second dataset (and even if you could, it might be so small you dont have power to replicate).

Therefore, my suggestion would be to do what you did, but for the SNPs significant for sample size "1", report the proportion that are more significant for sample size "2" (and in the same direction, as you already have done).

Ideally, the proportion would be high (and if you wished, you could test if it was significantly greater than 0.5).

Response: We appreciate the reviewer's suggestion on further improving the validation of the results. We followed the suggestion to revise our analysis. Specifically, we now define a variant is replicated if the same variant is also identified in the next larger study with a smaller p-value and the same direction of effect. We present the results in Table 1 (Table 1 in this letter). We also include the previous results in the letter for comparison purpose (Table 2 in this letter).

At FDR=0.05, we observed that 338 out of 385 (87.8%) variants by GhostKnockoff analysis of Kunkle et al. (2019) are also replicated in the analysis of Schwartzentruber et al. (2021); 447 out of 634 (70.5%) variants identified by GhostKnockoff analysis of Schwartzentruber et al. (2021) are also replicated in the proposed meta-analysis of all nine studies. At FDR=0.10, 370 out of 448 (82.6%) variants identified by GhostKnockoff analysis of Kunkle et al. (2019) are also replicated in the analysis of Schwartzentruber et al. (2021); 510 out of 724 (70.4%) variants identified by GhostKnockoff analysis of Schwartzentruber et al. (2021) are also replicated in the proposed meta-analysis of all nine studies. Both proportions remain higher than 0.5. The lower proportion for Schewartzentruber vs. the meta-analysis is likely due to the fact that the meta-analysis aggregates

many other studies with different design, where many p-values are unnecessarily monotone as more studies are included.

Overall, we conclude that the proposed GhostKnockoff (conditional test + FDR control), though systematically different from the conventional GWAS, is a valid approach to make reproducible genetic discoveries.

Table 1. Replication of variants identified by GhostKnockoff in larger studies. The analysis reflects the application of the proposed method to a subset of samples and the validation of the findings when we increase the sample size. We present the number of identified variants by applying GhostKnockoff to summary statistics from Kunkle et al. (2019), Schwartzentruber et al. (2021) (a study aggregating samples from Kunkle et al. (2019) and UK Biobank based on a proxy AD phenotype), and the proposed GhostKnockoff meta-analysis based on the optimal weights combining nine studies. A genetic variant is replicated if the same variant is also identified in the next larger study with a smaller p-value and the same direction of effect.

		Source of summary statistics		
		Kunkle et al. (2019)	Schwartzentruber et al. (2021)	Proposed meta-analysis of all nine studies
FDR=0.05	Total number of discoveries	385	634	764
	Number (proportion) of replicated discoveries	338 (87.8%)	447 (70.5%)	-
FDR=0.10	Total number of discoveries	448	724	935
	Number (proportion) of replicated discoveries	370 (82.6%)	510 (70.4%)	-

Previous Table 1: Replication of variants identified by GhostKnockoff in larger studies. The analysis reflects the application of the proposed method to a subset of samples and the validation of the findings when we increase the sample size. We present the number of identified variants by applying GhostKnockoff to summary statistics from Kunkle et al. (2019), Schwartzentruber et al. (2021) (a study aggregating samples from Kunkle et al. (2019) and UK Biobank based on a proxy AD phenotype), and the proposed GhostKnockoff meta-analysis based on the optimal weights combining nine studies. A genetic variant is replicated if the same variant is also identified in the next larger study with the same direction of effect.

		Source of summary statistics		
		Kunkle et al. (2019)	Schwartzentruber et al. (2021)	Proposed meta-analysis of all nine studies
FDR=0.05	Total number	385	634	764
	Number (proportion) of replicated discoveries	345 (89.6%)	525 (83%)	-
FDR=0.10	Total number	448	724	935
	Number (proportion) of replicated discoveries	378 (84.4%)	599 (82.7%)	-

2. While I was happy with the responses to my other comments (particularly my very technical ones), I was not happy with the response to R2, Point 3. The authors claim to have carefully read my paper (their response to my Point 12), and yet that paper exactly provides a way to do the PRS analysis R2 suggested (by allowing them to generate a pseudo independent dataset. As in, they can divide the summary statistics from all samples into pseudo training ss and pseudo test ss, then use the former to construct the PRS, and the latter to test...

Response: We would like to clarify that the current manuscript focuses on feature selection instead of prediction. Therefore, we consider the PRS comparison to be slightly outside the scope of the

current manuscript. However, we still curated additional individual-level datasets and performed analyses to address this valuable comment.

We first applied the proposed GhostKnockoff method to the summary statistics from Kunkle et al. (2019). We used variants identified by our GhostKnockoff to construct PRS and compared it against PRS obtained from the variants identified by the conventional approach. To make such a comparison, we curated individual-level data from the UK biobank, which consists of 388,051 unrelated participants independent of Kunkel et al. (2019) after pre-processing steps. We split the data into training and testing with an 80:20 ratio and stratification based on the phenotype labels. Based on the same number of top variants identified by GhostKnockoff (defined by the W statistics) and conventional GWAS (defined by p-values from Kunkle et al. 2019), we trained the PRS using the Lasso model based on 5-fold cross-validation. We compared the AUC using the test data and present the results in Figure 1 in this letter. To make robust comparisons, we report the result of mean AUC from 5 random train-test split.

We observed that the PRS constructed based on the results of GhostKnockoff is slightly higher than that constructed based on conventional approach. This is presumably due to the facts that GhostKnockoff tends to identify variants that carry independent information. However, the current improvement is not substantial (<0.01 in AUC). We think these preliminary results set the stage for an interesting discussion about the PRS based on knockoff inference which might be of great interest to the scientific community in the future. Should Reviewer 1 wish us to engage further in this discussion, we would be happy to do so, but at present believe that it may be outside the scope of our current work.

Figure 1. Comparison between AUC obtained from variants discovered from the conventional setup and variants discovered from the proposed knockoff framework.

Lastly, it seems from the response, that the editor has not yet responded to my Point 11.

Response: We have revised the paper to shorten the overview of methods in the Results section.

REVIEWERS' COMMENTS

Reviewer #1 (Remarks to the Author):

Thank you for the responses, I am satisfied.

Regarding 1 - thank you for the additional numbers, and for adding my suggestion.

Regarding 2 - I agree that PRS is not required. Thank you for the analyses; I believe these results are only in the response to reviewer (not also in the paper), which is absolutely fine.